# EXPLAINING BY IMITATING:
# UNDERSTANDING DECISIONS BY
# INTERPRETABLE POLICY LEARNING

**Alihan Hüyük**$^*$
University of Cambridge, UK
ah2075@cam.ac.uk

**Daniel Jarrett**$^*$
University of Cambridge, UK
daniel.jarrett@maths.cam.ac.uk

**Cem Tekin**
Bilkent University, Ankara, Turkey
cemtekin@ee.bilkent.edu.tr

**Mihaela van der Schaar**
University of Cambridge, UK
Cambridge Centre for AI in Medicine, UK
The Alan Turing Institute, UK; UCLA, USA
mv472@cam.ac.uk

## ABSTRACT

Understanding human behavior from observed data is critical for transparency and accountability in decision-making. Consider real-world settings such as healthcare, in which modeling a decision-maker's policy is challenging—with no access to underlying states, no knowledge of environment dynamics, and no allowance for live experimentation. We desire learning a data-driven representation of decision-making behavior that (1) inheres *transparency by design*, (2) accommodates *partial observability*, and (3) operates *completely offline*. To satisfy these key criteria, we propose a novel model-based Bayesian method for *interpretable policy learning* ("INTERPOLE") that jointly estimates an agent's (possibly biased) belief-update process together with their (possibly suboptimal) belief-action mapping. Through experiments on both simulated and real-world data for the problem of Alzheimer's disease diagnosis, we illustrate the potential of our approach as an investigative device for auditing, quantifying, and understanding human decision-making behavior.

## 1 INTRODUCTION

A principal challenge in modeling human behavior is in obtaining a transparent *understanding* of decision-making. In medical diagnosis, for instance, there is often significant regional and institutional variation in clinical practice [1], much of it the leading cause of rising healthcare costs [2]. The ability to quantify different decision processes is the first step towards a more systematic understanding of medical practice. Purely by observing demonstrated behavior, our principal objective is to answer the question: Under any given state of affairs, what actions are (more/less) likely to be taken, and why?

We address this challenge by setting our sights on three key criteria. First, we desire a method that is *transparent by design*. Specifically, a transparent description of behavior should locate the factors that contribute to individual decisions, in a language readily understood by domain experts [3, 4]. This will be clearer per our subsequent formalism, but we can already note some contrasts: Classical imitation learning—popularly by reduction to supervised classification—does not fit the bill, since black-box hidden states of RNNs are rarely amenable to meaningful interpretation. Similarly, apprenticeship learning algorithms—popularly through inverse reinforcement learning—do not satisfy either, since the high-level nature of reward mappings is not informative as to individual actions observed in the data. Rather than focusing purely on *replicating* actions (imitation learning) or on *matching* expert performance (apprenticeship learning), our chief pursuit lies in *understanding* demonstrated behavior.

Second, real-world environments such as healthcare are often *partially observable* in nature. This requires modeling the accumulation of information from entire sequences of past observations—an endeavor that is prima facie at odds with the goal of transparency. For instance, in a fully-observable setting, (model-free) behavioral cloning is arguably 'transparent' in providing simple mappings of states to actions; however, coping with partial observability using any form of recurrent function

---

$^*$Authors contributed equally

Table 1: *Comparison with Related Work.* INTERPOLE satisfies our key criteria of (1) transparency by design, (2) partial observability, and (3) offline learning, and makes no assumptions w.r.t. unbiasedness of beliefs or optimality of policies. Observations, beliefs, (optimal) q-values, actions, and policies are denoted $z$, $b$, $q_*$, $a$, and $\pi$; **bold** denotes learned quantities, *italics* are known (or queryable), and "†" denotes jointly-learned quantities.

| Approach | | Prototype | Overview | (1) | (2) | (3) | Beliefs | Policy |
|---|---|---|---|---|---|---|---|---|
| Imitation Learning | BC | [5] | $z \xrightarrow[\text{(supervised learning)}]{\pi(a\|z)} a$ | ✗∨✗ | | ✓ | (N/A) | No assumption |
| | GAIL | [6] | $z \xrightarrow[\text{(informed by } dynamics\text{)}]{\pi(a\|z)} a$ | ✗∨✗ | | ✗ | (N/A) | No assumption |
| | MB-IL | [7] | $z \xrightarrow[\text{(informed by } \textbf{environment model}\text{)}]{\pi(a\|z)} a$ | ✓ | ✗ | ✓ | (N/A) | No assumption |
| Apprenticeship Learning | IRL | [8] | $z \xrightarrow[\text{(reinforcement learning)}]{\textbf{rewards}, dynamics} q_* \xrightarrow{\text{(argmax)}} a$ | ✗ | ✗ | ✗ | (N/A) | Optimal |
| | PO-IRL | [9] | $z \xrightarrow[\text{(inference)}]{dynamics} b \xrightarrow[\text{(reinforcement learning)}]{\textbf{rewards}, dynamics} q_* \xrightarrow{\text{(argmax)}} a$ | ✗ | ✓ | ✗ | Unbiased | Optimal |
| | Off. PO-IRL | [10] | $z \xrightarrow[\text{(inference)}]{\textbf{dynamics}^\dagger} b \xrightarrow[\text{(reinforcement learning)}]{\textbf{rewards}^\dagger, \textbf{dynamics}^\dagger} q_* \xrightarrow{\text{(argmax)}} a$ | ✗ | ✓ | ✓ | Unbiased | Optimal |
| **INTERPOLE** | | **(Ours)** | $z \xrightarrow[\text{(inference)}]{\textbf{decision dynamics}^\dagger} b \xrightarrow{\pi(a\|b) \sim \textbf{decision boundaries}^\dagger} a$ | ✓ | ✓ | ✓ | No assumption | No assumption |

approximation immediately lands in black-box territory. Likewise, while (model-based) methods have been developed for robotic control, their transparency crucially hinges on fully-observable kinematics.

Finally, in realistic settings it is often impossible to experiment online—especially in high-stakes environments with real products and patients. The vast majority of recent work in (inverse) reinforcement learning has focused on games, simulations, and gym environments where access to live interaction is unrestricted. By contrast, in healthcare settings the environment dynamics are neither known a priori, nor estimable by repeated exploration. We want a data-driven representation of behavior that is learnable in a *completely offline* fashion, yet does not rely on knowing/modeling any true dynamics.

**Contributions** Our contributions are three-fold. First, we propose a model for interpretable policy learning ("INTERPOLE")—where sequential observations are aggregated through a decision agent's *decision dynamics* (viz. subjective belief-update process), and sequential actions are determined by the agent's *decision boundaries* (viz. probabilistic belief-action mapping). Second, we suggest a Bayesian learning algorithm for estimating the model, simultaneously satisfying the key criteria of transparency, partial observability, and offline learning. Third, through experiments on both simulated and real-world data for Alzheimer's disease diagnosis, we illustrate the potential of our method as an investigative device for auditing, quantifying, and understanding human decision-making behavior.

## 2 RELATED WORK

We seek to learn an *interpretable* parameterization of observed behavior to understand an agent's actions. Fundamentally, this contrasts with *imitation* learning (which seeks to best replicate demonstrated policies) and *apprenticeship* learning (which seeks to match some notion of performance).

**Imitation Learning** In fully-observable settings, behavior cloning (BC) readily reduces the imitation problem to one of supervised classification [5, 11–13]; i.e. actions are simply regressed on observations. While this can be extended to account for partial observability by parameterizing policies via recurrent function approximation [14], it immediately gives up on ease of interpretability per the black-box nature of RNN hidden states. A plethora of model-free techniques have recently been developed, which account for information in the rollout dynamics of the environment during policy learning (see e.g. [15–20])—most famously, generative adversarial imitation learning (GAIL) based on state-distribution matching [6, 21]. However, such methods require repeated online rollouts of intermediate policies during training, and also face the same black-box problem as BC in partially observable settings. Clearly in model-free imitation, it is difficult to admit both transparency and partial observability.

Specifically with an eye on explainability, Info-GAIL [22, 23] proposes an orthogonal notion of "interpretability" that hinges on clustering similar demonstrations to explain variations in behavior. However, as with GAIL it suffers from the need for live interaction for learning. Finally, several model-based techniques for imitation learning (MB-IL) have been studied in the domain of robotics. [24] consider kinematic models designed for robot dynamics, while [25] and [7] consider (non-)linear autoregressive exogenous models. However, such approaches invariably operate in fully-observable settings, and are restricted models hand-crafted for specific robotic applications under consideration.

**Apprenticeship Learning**   In subtle distinction to imitation learning, methods in apprenticeship learning assume the observed behavior is optimal with respect to some underlying reward function. Apprenticeship thus proceeds indirectly—often through inverse reinforcement learning (IRL) in order to infer a reward function, which (with appropriate optimization) generates learned behavior that matches the performance of the original—as measured by the rewards (see e.g. [8, 26–29]). These approaches have been variously extended to cope with partial observability (PO-IRL) [9, 30], to offline settings through off-policy evaluation [31–33], as well as to learned environment models [10].

However, a shortcoming of such methods is the requirement that the demonstrated policy in fact be *optimal* with respect to a true reward function that lies within an (often limited) hypothesis class under consideration—or is otherwise black-box in nature. Further, learning the true environment dynamics [10] corresponds to the requirement that policies be restricted to the class of functions that map from *unbiased* beliefs (cf. exact inference) into actions. Notably though, [34] considers both a form of suboptimality caused by time-inconsistent agents as well as biased beliefs. However, perhaps most importantly, due to the indirect, task-level nature of reward functions, inverse reinforcement learning is essentially opposed to our central goal of transparency—that is, in providing direct, action-level descriptions of behavior. In Section 5, we provide empirical evidence of this notion of interpretability.

**Towards INTERPOLE**   In contrast, we avoid making any assumptions as to either unbiasedness of beliefs or optimality of policies. After all, the former requires estimating (externally) "true" environment dynamics, and the latter requires specifying (objectively) "true" classes of reward functions—neither of which are necessary per our goal of transparently describing individual actions. Instead, INTERPOLE simply seeks the *most plausible explanation* in terms of (internal) decision dynamics and (subjective) decision boundaries. To the best of our knowledge, our work is the first to tackle all three key criteria—while making no assumptions on the generative process behind behaviors. Table 1 contextualizes our work, showing typical incarnations of related approaches and their graphical models.

Before continuing, we note that the separation between the internal dynamics of an agent and the external dynamics of the environment has been considered in several other works, though often for entirely different problem formulations. Most notably, [35] tackles the same policy learning problem as we do in online, fully-observable environments but for agent's with internal states that cannot be observed. They propose agent Markov models (AMMs) to model such environment-agent interactions. For problems other than policy learning, [36–38] also consider the subproblem of inferring an agent's internal dynamics; however, none of these works satisfy all three key criteria simultaneously as we do.

## 3   INTERPRETABLE POLICY LEARNING

We first introduce INTERPOLE's model of behavior, formalizing notions of decision dynamics and decision boundaries. In the next section, we suggest a Bayesian algorithm for model-learning from data.

**Problem Setup**   Consider a partially-observable decision-making environment in discrete time. At each step $t$, the agent takes action $a_t \in A$ and observes outcome $z_t \in Z$.[1] We have at our disposal an observed dataset of *demonstrations* $\mathcal{D} = \{(a_1^i, z_1^i, \ldots, a_{\tau_i}^i, z_{\tau_i}^i)\}_{i=1}^n$ by an agent, $\tau_i$ being the length of the $i$-th trajectory (we shall omit indices $i$ unless required). Denote by $h_t \doteq (a_1, z_1, \ldots, a_{t-1}, z_{t-1})$ the observed history at the beginning of step $t$, where $h_1 \doteq \emptyset$. Analogously, let $H_t \doteq (A \times Z)^{t-1}$ indicate the set of all possible histories at the start of step $t$, where $H_1 \doteq \{\emptyset\}$, and let $H \doteq \cup_{t=1}^{\infty} H_t$.

A proper policy $\pi$ is a mapping $\pi \in \Delta(A)^H$ from observed histories to action distributions, where $\pi(a|h)$ is the probability of taking action $a$ given $h$. We assume that $\mathcal{D}$ is generated by an agent acting according to some *behavioral policy* $\pi_b$. The problem we wish to tackle, then, is precisely how to obtain an interpretable parameterization of $\pi_b$. We proceed in two steps: First, we describe a parsimonious belief-update process for accumulating histories—which we term *decision dynamics*. Then, we take beliefs to actions via a probabilistic mapping—which gives rise to *decision boundaries*.

**Decision Dynamics**   We model belief-updates by way of an input-output hidden Markov model (IOHMM) identified by the tuple $(S, A, Z, T, O, b_1)$, with $S$ being the finite set of underlying states. $T \in \Delta(S)^{S \times A}$ denotes the transition function such that $T(s_{t+1}|s_t, a_t)$ gives the probability of transitioning into state $s_{t+1}$ upon action $a_t$ in state $s_t$, and $O \in \Delta(Z)^{A \times S}$ denotes the observation function such that $O(z_t|a_t, s_{t+1})$ gives the probability of observing $z_t$ after taking action $a_t$ and transitioning into state $s_{t+1}$. Finally, let beliefs $b_t \in \Delta(S)$ indicate the probability $b_t(s)$ that the

---

[1]While we take it here that $Z$ is finite, our method can easily be generalized to allow continuous observations.

environment exists in any state $s \in S$ at time $t$, and let $b_1$ give the initial state distribution. Note that—unlike in existing uses of the IOHMM formalism—these "probabilities" are for representing the thought process of *the human*, and may freely diverge from the actual mechanics of *the world*. To aggregate observed histories as beliefs, we identify $b_t(s)$ with $\mathbb{P}(s_t = s|h_t)$—an interpretation that leads to the recursive belief-update process (where in our problem, quantities $T, O, b_1$ are unknown):

$$b_{t+1}(s') \propto \sum_{s \in S} b_t(s)T(s'|s, a_t)O(z_t|a_t, s') \tag{1}$$

A key distinction bears emphasis: We do *not* require that this latter set of quantities correspond to (external) environment dynamics—and we do not obligate ourselves to recover any such notion of "true" parameters. To do so would imply the assumption that the agent in fact performs exactly *unbiased* inference on a perfectly known model of the environment, which is restrictive. It is also unnecessary, since our mandate is simply to model the (internal) mechanics of decision-making— which could well be generated from *possibly biased* beliefs or imperfectly known models of the world. In other words, our objective (see Equation 3) of simultaneously determining the most likely beliefs (cf. decision dynamics) and policies (cf. decision boundaries) is fundamentally more parsimonious.

**Decision Boundaries**   Given decision dynamics, a policy is then equivalently a map $\pi \in \Delta(A)^{\Delta(S)}$. Now, what is an interpretable parameterization? Consider the three-state example in Figure 1. We argue that a probabilistic parameterization that directly induces "decision regions" (cf. panel 1b) over the belief simplex is uniquely interpretable. For instance, strong beliefs that a patient has underlying mild cognitive impairment may map to the region where a specific follow-up test is promptly prescribed; this parameterization allows clearly locating such regions—as well as their boundaries.

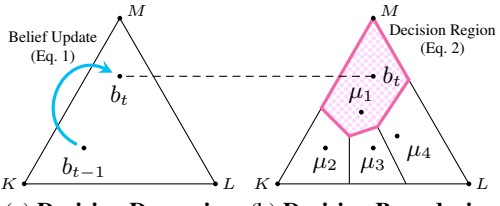

(a) **Decision Dynamics**   (b) **Decision Boundaries**

Figure 1: *The* INTERPOLE *Model.* Here, $S = \{K, L, M\}$ and $A = \{1, 2, 3, 4\}$. (a) Beliefs are updated recursively (Equation 1). (b) Actions are chosen with respect to relative locations of mean vectors (Equation 2).

Precisely, we parameterize policies in terms of $|A|$-many "mean" vectors that correspond to actions:

$$\pi(a|b) = e^{-\eta\|b-\mu_a\|^2} / \sum_{a' \in A} e^{-\eta\|b-\mu_{a'}\|^2} , \quad \sum_{s \in S} \mu_a(s) = 1 \tag{2}$$

where $\eta \geq 0$ is the inverse temperature, $\|\cdot\|$ the $\ell_2$-norm, and $\mu_a \in \mathbb{R}^{|S|}$ the mean vector corresponding to action $a \in A$. Intuitively, mean vectors induce decision boundaries (and decision regions) over the belief space $\Delta(S)$: At any time, the action whose corresponding mean is closest to the current belief is most likely to be chosen. In particular, lines that are equidistant to the means of any pair of actions form decision boundaries between them. The inverse temperature controls the transitions between such boundaries: A larger $\eta$ captures more deterministic behavior (i.e. more "abrupt" transitions), whereas a smaller $\eta$ captures more stochastic behavior (i.e. "smoother" transitions). Note that the case of $\eta = 0$ recovers policies that are uniformly random, and $\eta \to \infty$ recovers argmax policies.

A second distinction is due: The exponentiated form of Equation 2 should not be confused with typical Boltzmann [27] or MaxEnt [39] policies common in RL: These are indirect parameterizations via optimal/soft $q$-values, which themselves require approximate solutions to optimization problems; as we shall see in our experiments, the quality of learned policies suffers as a result. Further, using $q$-values would imply the assumption that the agent in fact behaves *optimally* w.r.t. an (objectively) "true" class of reward functions—e.g. linear—which is restrictive. It is also unnecessary, as our mandate is simply to capture their (subjective) tendencies toward different actions—which are generated from *possibly suboptimal* policies. In contrast, by directly partitioning the belief simplex into probabilistic "decision regions", INTERPOLE's mean-vector representation can be immediately explained and understood.

**Learning Objective**   In a nutshell, our objective is to identify the most likely parameterizations $T$, $O, b_1$ for decision dynamics as well as $\eta, \{\mu_a\}_{a \in A}$ for decision boundaries, given the observed data:

$$\begin{array}{l} \text{Given: } \mathcal{D}, S, A, Z \\ \text{Determine: } T, O, b_1, \eta, \{\mu_a\}_{a \in A} \end{array} \tag{3}$$

Next, we propose a Bayesian algorithm that finds the maximum a posteriori (MAP) estimate of these quantities. Figure 2 illustrates the problem setup.

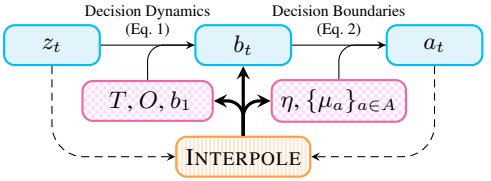

Figure 2: *The* INTERPOLE *Objective.* Inputs (demonstration data) are fed into INTERPOLE through dashed lines, and outputs (estimates) are issued in bold lines. (Beliefs $b_t$ can then be computed via a forward pass).

## 4 BAYESIAN INTERPRETABLE POLICY LEARNING

Denote with $\theta \doteq (T, O, b_1, \eta, \{\mu_a\}_{a \in A})$ the set of parameters to be determined, and let $\theta$ be drawn from some prior $\mathbb{P}(\theta)$. In addition, denote with $\bar{\mathcal{D}} = \{(s_1^i, \ldots, s_{\tau_i+1}^i)\}_{i=1}^n$ the set of underlying (unobserved) state trajectories, such that $\mathcal{D} \cup \bar{\mathcal{D}}$ gives the *complete* (fully-observed) dataset. Then the complete likelihood—of the unknown parameters $\theta$ with respect to $\mathcal{D} \cup \bar{\mathcal{D}}$—is given by the following:

$$\mathbb{P}(\mathcal{D}, \bar{\mathcal{D}}|\theta) = \underbrace{\prod_{i=1}^n \prod_{t=1}^\tau \pi(a_t|b_t[T, O, b_1, h_{t-1}])}_{\text{action likelihoods}} \times \underbrace{\prod_{i=1}^n b_1(s_1) \prod_{t=1}^\tau T(s_{t+1}|s_t, a_t) O(z_t|a_t, s_{t+1})}_{\text{observation likelihoods}} \quad (4)$$

where $\pi(\cdot|\cdot)$ is described by $\eta$ and $\{\mu_a\}_{a \in A}$, and each $b_t[\cdot]$ is a function of $T$, $O$, $b_1$, and $h_{t-1}$ (Equation 1). Since we do not have access to $\bar{\mathcal{D}}$, we propose an expectation-maximization (EM)-like algorithm for maximizing the posterior $\mathbb{P}(\theta|\mathcal{D}) = \mathbb{P}(\mathcal{D}|\theta)\mathbb{P}(\theta)/\int \mathbb{P}(\mathcal{D}|\theta)\mathrm{d}\mathbb{P}(\theta)$ over the parameters:

**Bayesian Learning** Given an initial estimate $\hat{\theta}^0$, we iteratively improve the estimate by performing the following steps at each iteration $k$:

- "*E-step*": Compute the expected log-likelihood of the model parameters $\theta$ given the previous parameter estimate $\hat{\theta}^{k-1}$, as follows:

$$\begin{aligned} Q(\theta; \hat{\theta}^{k-1}) &\doteq \mathbb{E}_{\bar{\mathcal{D}}|\mathcal{D}, \hat{\theta}^{k-1}}[\log \mathbb{P}(\mathcal{D}, \bar{\mathcal{D}}|\theta)] \\ &= \sum_{\bar{\mathcal{D}}} \log \mathbb{P}(\mathcal{D}, \bar{\mathcal{D}}|\theta) \mathbb{P}(\bar{\mathcal{D}}|\mathcal{D}, \hat{\theta}^{k-1}) \end{aligned} \quad (5)$$

where we compute the necessary marginalizations of joint distribution $\mathbb{P}(\bar{\mathcal{D}}|\mathcal{D}, \hat{\theta}^{k-1})$ by way of a forward-backward procedure (detailed procedure given in Appendix A.1).

---

**Algorithm 1** Bayesian INTERPOLE

1: **Parameters**: learning rate $w \in \mathbb{R}_+$
2: **Input**: dataset $\mathcal{D} = \{h_{\tau_i+1}^i\}_{i=1}^n$, prior $\mathbb{P}(\theta)$
3: Sample $\hat{\theta}^0$ from $\mathbb{P}(\theta)$
4: **for** $k = 1, 2, \ldots$ **do**
5:    Compute $\mathbb{P}(\bar{\mathcal{D}}|\mathcal{D}, \hat{\theta}^{k-1})$      ▷Appendix A.1
6:    Compute $\nabla_\theta Q(\theta; \hat{\theta}^{k-1})$ at $\hat{\theta}^{k-1}$ ▷Appendix A.2
7:    $\hat{\theta}^k \leftarrow \hat{\theta}^{k-1}$
        $+ w[\nabla_\theta Q(\theta; \hat{\theta}^{k-1}) + \nabla_\theta \log \mathbb{P}(\theta)]_{\theta = \hat{\theta}^{k-1}}$
8: **while** (6)
9: $\hat{\theta} \leftarrow \hat{\theta}^{k-1}$
10: **Output**: MAP estim. $\hat{\theta} \doteq (\hat{T}, \hat{O}, \hat{b}_1, \hat{\eta}, \{\hat{\mu}_a\}_{a \in A})$

---

- "*M-step*": Compute a new estimate $\hat{\theta}^k$ that improves the expected log-posterior—that is, such that:

$$Q(\hat{\theta}^k; \hat{\theta}^{k-1}) + \log \mathbb{P}(\hat{\theta}^k) > Q(\hat{\theta}^{k-1}; \hat{\theta}^{k-1}) + \log \mathbb{P}(\hat{\theta}^{k-1}) \quad (6)$$

subject to appropriate non-negativity and normalization constraints on parameters, which can be achieved via gradient-based methods (detailed procedure given in Appendix A.2). We stop when it becomes no longer possible to obtain a new estimate further improving the expected log-posterior —that is, when the "M-step" cannot be performed. Algorithm 1 summarizes this learning procedure.

**Explaining by Imitating** Recall our original mandate—to give the *most plausible explanation* for behavior. Two questions can be asked about our proposal to "explain by imitating"—to which we now have precise answers: One concerns explainability, and the other concerns directness of explanations.

First, what constitutes the "most plausible explanation" of behavior? Now, INTERPOLE identifies this as the *most likely parameterization* of that behavior using a state-based model for beliefs and policies—but otherwise with no further assumptions. In particular, we are only positing that modeling beliefs over states helps provide an interpretable description of how an *agent* reasons (which we do have ample evidence for[2])—but we are not assuming that the *environment* itself takes the form of a state-based model (which is an entirely different claim). Mathematically, the complete likelihood (Equation 4) highlights the difference between decision dynamics (which help explain the agent's behavior) and "true" environment dynamics (which we do not care about). The latter are independent of the agent, and learning them would have involved just the observation likelihoods alone. In contrast, by *jointly* estimating $T, O, b_1$ with $\eta, \{\mu_a\}_{a \in A}$ according to both the observation- and action-likelihoods, we are learning the decision dynamics—which in general need not coincide with the environment, but which offer the most plausible explanation of how the agent effectively reasons.

The second question is about *directness*: Given the popularity of the IRL paradigm, could we have simply used an (indirect) reward parameterization, instead of our (direct) mean-vector parameterization? As it turns out, in addition to the "immediate" interpretability of direct, action-level representations,

---

[2]In healthcare, diseases are often modeled in terms of states, and beliefs over disease states are eminently transparent factors that medical practitioners (i.e. domain experts) readily comprehend and reason about [40, 41].

it comes with an extra perk w.r.t. computability: While it is (mathematically) possible to formulate a similar learning problem swapping out $\mu$ for rewards, in practice it is (computationally) intractable to perform in our setting. The precise difficulty lies in differentiating through quantities $\pi(a_t|b_t)$—which in turn depend on beliefs and dynamics—in the action-likelihoods (proofs located in Appendix B):

**Proposition 1** (*Differentiability with q-Parameterizations*) Consider softmax policies parameterized by $q$-values from a reward function, such that $\pi(a|b) = e^{q_*(b,a)} / \sum_{a'} e^{q_*(b,a')}$ in lieu of Equation 2. Then differentiating through $\log \pi(a_t|b_t)$ terms with respect to unknown parameters $\theta$ is *intractable*.

In contrast, INTERPOLE avoids ever needing to solve any "forward" problem at all (and therefore does not require resorting to costly—and approximate—sampling-based workarounds) for learning:

**Proposition 2** (*Differentiability with $\mu$-Parameterizations*) Consider the mean-vector policy parameterization proposed in Equation 2. Differentiation through the $\log \pi(a_t|b_t)$ terms with respect to the unknown parameters $\theta$ is easily and automatically performed using *backpropagation* through time.

## 5 ILLUSTRATIVE EXAMPLES

Three aspects of INTERPOLE deserve empirical demonstration, and we shall highlight them in turn:

- *Interpretability*: First, we illustrate the usefulness of our method in providing transparent explanations of behavior. This is our primary objective here——of explaining by imitating.
- *Accuracy*: Second, we demonstrate that the faithfulness of learned policies is not given up for transparency. This shows that accuracy and interpretability are not necessarily opposed.
- *Subjectivity*: Third, we show INTERPOLE correctly recovers underlying explanations for behavior——even if the agent is biased. This sets us apart from other state-based algorithms.

In order to do so, we show archetypical examples to exercise our framework, using both simulated and real-world experiments in the context of *disease diagnosis*. State-based reasoning is prevalent in research and practice: three states in progressive clinical dementia [42, 43], preterminal cancer screening [44, 45], or even—as recently shown—for cystic fibrosis [46] and pulmonary disease [47].

**Decision Environments** For our real-world setting, we consider the diagnostic patterns for 1,737 patients during sequences of 6-monthly visits in the Alzheimer's Disease Neuroimaging Initiative [48] database (**ADNI**). The state space consists of normal functioning ("NL"), mild cognitive impairment ("MCI"), and dementia. For the action space, we consider the decision problem of ordering vs. not ordering an MRI test, which—while often informative of Alzheimer's—is financially costly. MRI outcomes are categorized according to hippocampal volume: {"avg", "above avg", "below avg", "not ordered"}; separately, the cognitive dementia rating-sum of boxes ("CDR-SB") result— which is always measured—is categorized as: {"normal", "questionable impairment", "mild/severe dementia"} [42]. In total, the observation space therefore consists of the 12 combinations of outcomes.

We also consider a simulated setting to better validate performance. For this we employ a diagnostic environment (**DIAG**) in the form of an IOHMM with certain (true) parameters $T^{\text{true}}, O^{\text{true}}, b_1^{\text{true}}$. Patients fall within diseased ($s_+$) and healthy ($s_-$) states, and vital-sign measurements available at every step are classified within positive ($z_+$) and negative ($z_-$) outcomes. For the action space, we consider the decision of continuing to monitor a patient ($a_=$), or stopping and declaring a final diagnosis—and if so, a diseased ($a_+$) or healthy ($a_-$) declaration. If we assume agents have perfect knowledge of the true environment, then this setup is similar to the classic "tiger problem" for optimal stopping [49]. Lastly, we also consider a (more realistic) variant of DIAG where the agent's behavior is instead generated by biased beliefs due to incorrect knowledge $T, O, b_1 \neq T^{\text{true}}, O^{\text{true}}, b_1^{\text{true}}$ of the environment (**BIAS**). Importantly, this generates a testable version of real-life settings where decision-makers' (subjective) beliefs often fail to coincide with (objective) probabilities in the world.

**Benchmark Algorithms** Where appropriate, we compare INTERPOLE against the following benchmarks: imitation by behavioral cloning [5] using RNNs for partial observability (**R-BC**); Bayesian IRL on POMDPs [9] equipped with a learned environment model (**PO-IRL**); a fully-offline counterpart [10] of Bayesian IRL (**Off. PO-IRL**); and an adaptation of model-based imitation learning [7] to partially-observable settings, with a learned IOHMM as the model (**PO-MB-IL**). Algorithms requiring learned models for interaction are given IOHMMs estimated using conventional methods [50]. Further information on environments and benchmark implementations is found in Appendix C.

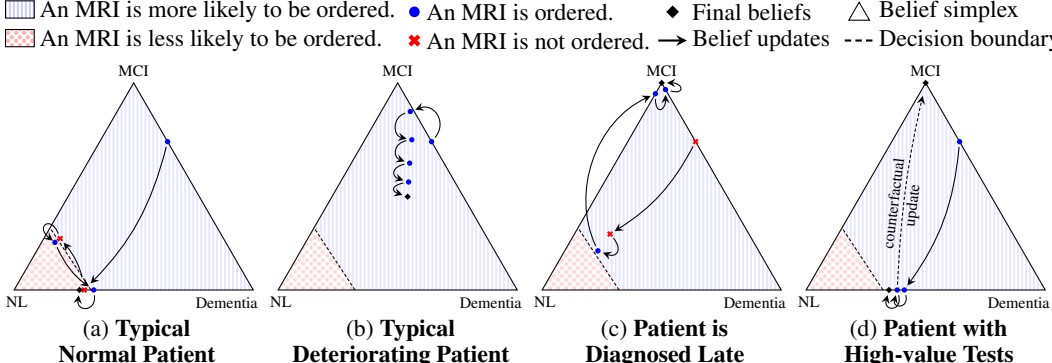

Figure 3: *Decision Trajectories*. Examples of real patients, including: (a) A typical normally-functioning patient, where the decision-maker's beliefs remain mostly on the decision boundary. (b) A typical patient who is believed to be deteriorating towards dementia. (c) A patient who—apparently—could have been diagnosed much earlier than they actually were. (d) A patient with a (seemingly redundant) MRI test that is actually highly informative.

**Interpretability** First, we direct attention to the potential utility of INTERPOLE as an *investigative device* for auditing and quantifying individual decisions. Specifically, modeling the evolution of an agent's beliefs provides a concrete basis for analyzing the corresponding sequence of actions taken:

- *Explaining Trajectories*. Figure 3 shows examples of such decision trajectories for four real ADNI patients. Each vertex of the belief simplex corresponds to one of the three stable diagnoses, and each point in the simplex corresponds to a unique belief (i.e. probability distribution). The closer the point is to a vertex (i.e. state), the higher the probability assigned to that state. For instance, if the belief is located exactly in the middle of the simplex (i.e. equidistant from all vertices), then all states are believed to be equally likely. If the belief is located exactly on a vertex (e.g. directly on top of MCI), then this corresponds to an absolutely certainty of MCI being the underlying state. Patients (a) and (b) are "typical" patients who fit well to the overall learned policy. The former is a normally-functioning patient believed to remain around the decision boundary in all visits except the first; appropriately, they are ordered an MRI during approximately half of their visits. The latter is believed to be deteriorating from MCI towards dementia, hence prescribed an MRI in all visits.

- *Identifying Belated Diagnoses*. In many diseases, early diagnosis is paramount [51]. INTERPOLE allows detecting patients who appear to have been diagnosed significantly later than they should have. Patient (c) was ordered an MRI in neither of their first two visits—despite the fact that the "typical" policy would have strongly recommended one. At a third visit, the MRI that was finally ordered led to near-certainty of cognitive impairment—but this could have been known 12 months earlier! In fact, among all ADNI patients in the database, 6.5% were subject to this apparent pattern of "belatedness", where a late MRI is immediately followed by a jump to near-certain deterioration.

- *Quantifying Value of Information*. Patient (d) highlights how INTERPOLE can be used to quantify the value of a test in terms of its information gain. While the patient was ordered an MRI in all of their visits, it may appear (on the surface) that the third and final MRIs were redundant—since they had little apparent affect on beliefs. However, this is only true for the *factual* belief update that occurred according to the MRI outcome that was actually observed. Having access to an estimated model of how beliefs are updated in the form of decision dynamics, we can also compute *counterfactual* belief updates—that is belief updates that could have occurred if the MRI outcome in question were to be different. In the particular case of patient (d), the tests were in fact highly informative, since (as it happened) the patient's CDR-SB scores were suggestive of impairment, and (in the counterfactual) the doctor's beliefs could have potentially leapt drastically towards MCI. On the other hand, among all MRIs ordered for ADNI patients, 19% may indeed have been unnecessary (i.e. triggering apparently insignificant belief-updates both factually as well as counterfactually).

Evaluating Interpretability through Clinician Surveys

To cement the argument for interpretability, we evaluated INTERPOLE by consulting nine clinicians from four different countries (United States, United Kingdom, the Netherlands, and China) for feedback. We focused on evaluating two aspects of interpretability regarding our method:

- *Decision Dynamics*: Whether the proposed representation of (possibly subjective) belief trajectories are preferable to raw action-observation trajectories—that is, whether decision dynamics are a transparent way of modeling how information is aggregated by decision-makers.

- *Decision Boundaries*: Whether the proposed representation of (possibly suboptimal) decision boudnaries are a more transparent way of describing policies, compared with the representation of reward functions (which is the conventional approach in the policy learning literature).

For the first aspect, we presented to the participating clinicians the medical history of an example patient from ADNI represented in three ways using: only the most recent action-observation, the complete action-observation trajectory, as well as the belief trajectory as recovered by INTERPOLE. **Result**: All nine clinicians preferred the belief trajectories over action-observation trajectories.

For the second aspect, we showed them the policies learned from ADNI by both Off. PO-IRL and INTERPOLE, which parameterize policies in terms of reward functions and decision boundaries respectively. **Result**: Seven out of nine clinicians preferred the representation in terms of decision boundaries over that offered by reward functions. Further details can be found in Appendix D.

**Accuracy** Now, a reasonable question is whether such explainability comes at a cost: By learning an interpretable policy, do we sacrifice any accuracy? To be precise, we can ask the following questions:

- Is the *belief-update process* the same? For this, the appropriate metric is the discrepancy with respect to the sequence of beliefs—which we take to be $\sum_t D_{\mathrm{KL}}(b_t \| \hat{b}_t)$ (**Belief Mismatch**).

- Is the *belief-action* mapping the same? Our metric is the discrepancy with respect to the policy distribution itself—which we take to be $\sum_t D_{\mathrm{KL}}(\pi_{\mathrm{b}}(\cdot|b_t) \| \hat{\pi}(\cdot|\hat{b}_t))$ (**Policy Mismatch**).

- Is the *effective behavior* the same? Here, the metrics are those measuring the discrepancy with respect to ground-truth actions observed (**Action-Matching**) for ADNI, and differences in stopping (**Stopping Time Error**) for DIAG.

Note that the action-matching and stopping time errors evaluate the quality of learned models in *imitating* per se, whereas belief mismatch and policy mismatch evaluate their quality in *explaining*.[3]

Table 2: *Performance Comparison in ADNI.* INTERPOLE is best/second-best for action-matching metrics.

| Algorithm | Calibration (Brier Score) | Area under ROC Curve | Area under PR Curve |
|---|---|---|---|
| R-BC | $0.18 \pm 0.05$ | $0.61 \pm 0.05$ | $0.81 \pm 0.08$ |
| PO-MB-IL | $0.19 \pm 0.07$ | $0.54 \pm 0.07$ | $0.79 \pm 0.11$ |
| PO-IRL | $0.23 \pm 0.01$ | $0.51 \pm 0.07$ | $0.78 \pm 0.09$ |
| Off. PO-IRL | $0.24 \pm 0.01$ | $0.54 \pm 0.05$ | $0.79 \pm 0.09$ |
| INTERPOLE | $0.17 \pm 0.05$ | $0.60 \pm 0.04$ | $0.81 \pm 0.09$ |

Table 3: *Performance Comparison in DIAG.* INTERPOLE is best. Belief mismatch is n/a to R-BC. $^{\dagger} \times 10^{-3}$

| Algorithm | Belief Mismatch$^{\dagger}$ | Policy Mismatch$^{\dagger}$ | Stopping Time Error |
|---|---|---|---|
| R-BC | – | $6.7 \pm 1.4$ | $5.91 \pm 1.29$ |
| PO-MB-IL | $42.7 \pm 34.0$ | $47.9 \pm 18.9$ | $6.34 \pm 1.48$ |
| PO-IRL | $42.7 \pm 34.0$ | $62.1 \pm 29.1$ | $6.41 \pm 1.80$ |
| Off. PO-IRL | $26.1 \pm 5.6$ | $2.1 \pm 0.2$ | $5.42 \pm 0.69$ |
| INTERPOLE | $0.6 \pm 0.1$ | $0.8 \pm 0.2$ | $5.38 \pm 1.14$ |

The results are revealing, if not necessarily surprising. To begin, we observe for the ADNI setting in Table 2 that INTERPOLE performs first- or second-best across all three action-matching based metrics; where it comes second, it does so only by a small margin to R-BC (bearing in mind that R-BC is specifically optimized for nothing but action-matching). Similarly for the DIAG setting, we observe in Table 3 that INTERPOLE performs the best in terms of stopping-time error. In other words, it appears that little—if any—imitation accuracy is lost by using INTERPOLE as the model.

Perhaps more interestingly, we also see in Table 3 that the quality of internal explanations is superior—in terms of both belief mismatch and policy mismatch. In particular, even though the comparators PO-MB-IL, PO-IRL, and Off. PO-IRL are able to map decisions through beliefs, they inherit the conventional approach of attempting to estimate *true* environment dynamics, which is unnecessary—and possibly detrimental—if the goal is simply to find the *most likely* explanation of behavior. Notably, while the difference in imitation quality among the various benchmarks is not tremendous, with respect to explanation quality the gap is significant—where INTERPOLE has great advantage.

**Subjectivity** Most significantly, we now show that INTERPOLE correctly recovers the underlying explanations, even if—or perhaps *especially* if—the agent is driven by subjective reasoning (i.e. with biased beliefs). This aspect sets INTERPOLE firmly apart from the alternative state-based techniques.

---

[3] Belief/policy mismatch are not applicable to ADNI since we have no access to ground-truth beliefs/policies.

Consider the BIAS environment: Here the true environment dynamics are unchanged from DIAG, but no longer coincide with the agent's decision dynamics. Specifically, we let the behavioral policy be generated using erroneous parameters $O(z_-|a_=, s_+) < O^{\text{true}}(z_-|a_=, s_+)$; that is, the doctor now incorrectly believes the test to have a smaller *false-negative rate* than it does in reality —thus biasing their beliefs regarding patient states.

Now suppose we wish to recover the decision boundary from the demonstrations—that is, at what confidence threshold does a doctor commit to a healthy diagnosis? Figure 4 shows that both INTERPOLE and PO-MB-IL appear to recover the correct *effective behavior*: Starting from a neutral prior, doctors tend to stop and issue a "healthy" diagnosis if the first two observations return negative signals. Importantly, INTERPOLE also correctly locates the confidence target $\sim 90\%$. On the other hand, PO-MB-IL—which first attempts to estimate the environment's true parameters—ends up learning a policy on the basis of miscalibrated beliefs, thereby incorrectly explaining the same effective behavior with a lower confidence target $\sim 70\%$. Finally, through similarly benchmarked metrics, Table 4 further confirms INTERPOLE's advantage in providing the (correct) explanations.

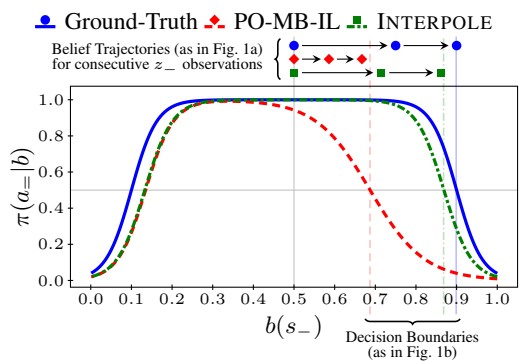

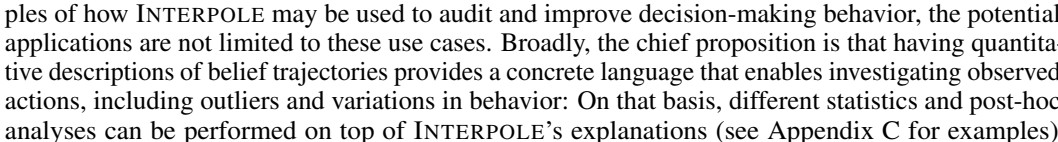

Figure 4: *Explaining Subjective Behavior*. Markers show the evolution of beliefs that explain ground-truth and learned policies in BIAS, for an example scenario where two consecutive negative $(z_-)$ observations are made. While all policies display similar effective behavior, only INTERPOLE correctly identifies the ground-truth decision boundary. This underscores the significance of distinguishing between decision dynamics (which help explain an agent's behavior) and "true" dynamics (which we do not care about).

Table 4: *Performance Comparison in BIAS*. INTERPOLE is best. Belief mismatch is n/a to R-BC. $^\dagger \times 10^{-3}$

| Algorithm | Belief Mismatch$^\dagger$ | Policy Mismatch$^\dagger$ | Stopping Time Error |
|---|---|---|---|
| R-BC | – | $191.8 \pm 78.8$ | $2.22 \pm 0.45$ |
| PO-MB-IL | $83.2 \pm 3.03$ | $125.4 \pm 18.4$ | $3.80 \pm 0.19$ |
| PO-IRL | $83.2 \pm 3.03$ | $144.0 \pm 29.4$ | $3.46 \pm 0.22$ |
| Off. PO-IRL | $84.9 \pm 8.28$ | $62.8 \pm 7.5$ | $3.12 \pm 0.41$ |
| INTERPOLE | $3.8 \pm 0.34$ | $1.8 \pm 0.6$ | $2.14 \pm 0.51$ |

## 6  DISCUSSION

Three points deserve brief comment in closing. First, while we gave prominence to several examples of how INTERPOLE may be used to audit and improve decision-making behavior, the potential applications are not limited to these use cases. Broadly, the chief proposition is that having quantitative descriptions of belief trajectories provides a concrete language that enables investigating observed actions, including outliers and variations in behavior: On that basis, different statistics and post-hoc analyses can be performed on top of INTERPOLE's explanations (see Appendix C for examples).

Second, a reasonable question is whether or not it is reasonable to assume access to the state space. For this, allow us to reiterate a subtle distinction. There may be some "ground-truth" external state space that is arbitrarily complex or even impossible to discover, but—as explained—we are not interested in modeling this. Then, there is the *internal* state space that an agent uses to reason about decisions, which is what we are interested in. In this sense, it is certainly reasonable to assume access to the state space, which is often very clear from medical literature [42–47]. Since our goal is to obtain interpretable representations of decision, it is therefore reasonable to cater precisely to these accepted state spaces that doctors can most readily reason with. Describing behavior in terms of beliefs over these (already well-understood) states is one of the main contributors to the interpretability of our method.

Finally, it is crucial to keep in mind that INTERPOLE does not claim to identify the *real* intentions of an agent: humans are complex, and rationality is—of course—bounded. What it does do, is to provide an interpretable explanation of how an agent is *effectively* behaving, which——as we have seen for diagnosis of ADNI patients——offers a yardstick by which to assess and compare trajectories and subgroups. In particular, INTERPOLE achieves this while adhering to our key criteria for healthcare settings, and without imposing assumptions of unbiasedness or optimality on behavioral policies.

### ACKNOWLEDGMENTS

This work was supported by the US Office of Naval Research (ONR) and Alzheimer's Research UK (ARUK). We thank the clinicians who participated in our survey, the reviewers for their valuable feedback, and the Alzheimer's Disease Neuroimaging Initiative for providing the ADNI dataset.

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

# A DETAILS OF THE ALGORITHM

## A.1 FORWARD-BACKWARD PROCEDURE

We compute the necessary marginalizations of the joint distribution $\mathbb{P}(\bar{\mathcal{D}}|\mathcal{D}, \hat{\theta})$ using the forward-backward algorithm. Letting $\boldsymbol{x}_{t:t'} = \{x_t, x_{t+1}, \ldots, x_{t'}\}$ for any time-indexed quantity $x_t$, the forward messages are defined as $\alpha_t(s) = \mathbb{P}(s_t = s, \boldsymbol{a}_{1:t-1}, \boldsymbol{z}_{1:t-1}|\hat{\theta})$, which can be computed dynamically as

$$
\begin{aligned}
\alpha_{t+1}(s') &= \mathbb{P}(s_{t+1} = s', \boldsymbol{a}_{1:t}, \boldsymbol{z}_{1:t}|\hat{\theta}) \\
&= \sum_{s \in S} \mathbb{P}(s_t = s, \boldsymbol{a}_{1:t-1}, \boldsymbol{z}_{1:t-1}|\hat{\theta})\mathbb{P}(s_{t+1} = s', a_t, z_t|s_t = s, \boldsymbol{a}_{1:t-1}, \boldsymbol{z}_{1:t-1}, \hat{\theta}) \\
&= \sum_{s \in S} \alpha_t(s)\hat{\pi}(a_t|b_t)\hat{T}(s'|s, a_t)\hat{O}(z_t|a_t, s') \\
&\propto \sum_{s \in S} \alpha_t(s)\hat{T}(s'|s, a_t)\hat{O}(z_t|a_t, s')
\end{aligned}
$$

with initial case $\alpha_1(s) = \mathbb{P}(s_1 = s) = b_1(s)$. The backward messages are defined as $\beta_t(s) = \mathbb{P}(\boldsymbol{a}_{t:\tau}, \boldsymbol{z}_{t:\tau}|s_t = s, \boldsymbol{a}_{1:t-1}, \boldsymbol{z}_{1:t-1}, \hat{\theta})$, which can also be computed dynamically as

$$
\begin{aligned}
\beta_t(s) &= \mathbb{P}(\boldsymbol{a}_{t:\tau}, \boldsymbol{z}_{t:\tau}|s_t = s, \boldsymbol{a}_{1:t-1}, \boldsymbol{z}_{1:t-1}, \hat{\theta}) \\
&= \sum_{s' \in S} \mathbb{P}(s_{t+1} = s', a_t, z_t|s_t = s, \boldsymbol{a}_{1:t-1}, \boldsymbol{z}_{1:t-1}, \hat{\theta})\mathbb{P}(\boldsymbol{a}_{t+1:\tau}, \boldsymbol{z}_{t+1:\tau}|s_{t+1} = s', \boldsymbol{a}_{1:t}, \boldsymbol{z}_{1:t}, \hat{\theta}) \\
&= \sum_{s' \in S} \hat{\pi}(a_t|b_t)\hat{T}(s'|s, a_t)\hat{O}(z_t|a_t, s')\beta_{t+1}(s') \\
&\propto \sum_{s' \in S} \hat{T}(s'|s, a_t)\hat{O}(z_t|a_t, s')\beta_{t+1}(s')
\end{aligned}
$$

with initial case $\beta_{\tau+1}(s) = \mathbb{P}(\emptyset|s_{\tau+1} = s, \boldsymbol{a}_{1:\tau}, \boldsymbol{z}_{1:\tau}, \hat{\theta}) = 1$.

Then, the marginal probability of being in state $s$ at time $t$ given the dataset $\mathcal{D}$ and the estimate $\hat{\theta}$ can be computed as

$$
\begin{aligned}
\gamma_t(s) &= \mathbb{P}(s_t = s|\mathcal{D}, \hat{\theta}) \\
&= \mathbb{P}(s_t = s|\boldsymbol{a}_{1:\tau}, \boldsymbol{z}_{1:\tau}, \hat{\theta}) \\
&\propto \mathbb{P}(s_t = s, \boldsymbol{a}_{1:\tau}, \boldsymbol{z}_{1:\tau}|\hat{\theta}) \\
&= \alpha_t(s)\beta(s)
\end{aligned}
$$

and similarly, the marginal probability of transitioning from state $s$ to state $s'$ at the end of time $t$ given the dataset $\mathcal{D}$ and the estimate $\hat{\theta}$ can be computed as

$$
\begin{aligned}
\xi_t(s, s') &= \mathbb{P}(s_t = s, s_{t+1} = s'|\mathcal{D}, \hat{\theta}) \\
&\propto \mathbb{P}(s_t = s, s_{t+1} = s', \boldsymbol{a}_{1:\tau}, \boldsymbol{z}_{1:\tau}|\hat{\theta}) \\
&= \mathbb{P}(s_t = s, \boldsymbol{a}_{1:t-1}, \boldsymbol{z}_{1:t-1}|\hat{\theta})\mathbb{P}(s_{t+1} = s', a_t, z_t|s_t = s, \boldsymbol{a}_{1:t-1}, \boldsymbol{z}_{1:t-1}, \hat{\theta}) \\
&\qquad\qquad\qquad\qquad\qquad \times \mathbb{P}(\boldsymbol{a}_{t+1:\tau}, \boldsymbol{z}_{t+1:\tau}|s_{t+1} = s', \boldsymbol{a}_{1:t}, \boldsymbol{z}_{1:t}, \hat{\theta}) \\
&= \alpha_t(s)\hat{\pi}(a_t|b_t)\hat{T}(s'|s, a)\hat{O}(z|a, s')\beta_{t+1}(s') \\
&\propto \alpha_t(s)\hat{T}(s'|s, a)\hat{O}(z|a, s')\beta_{t+1}(s') \ .
\end{aligned}
$$

## A.2 GRADIENT-ASCENT PROCEDURE

Taking the gradient of the expected log-likelihood $Q(\theta; \hat{\theta})$ in (5) with respect to the unknown parameters $\theta = (T, O, b_1, \eta, \mu_{a \in A})$ first requires computing the Jacobian matrix $\nabla_{b_t} b_{t'}$ for $1 \leq$

$t < t' \leq \tau$, where $(\nabla_b b')_{ij} = \partial b'(i)/\partial b(j)$ for $i, j \in S$. This can be achieved dynamically as $\nabla_{b_t} b_{t'} = \nabla_{b_{t+1}} b_{t'} \nabla_{b_t} b_{t+1}$ with initial case $\nabla_{b_{t'}} b_{t'} = I$, where

$$
\begin{aligned}
(\nabla_{b_t} b_{t+1})_{ij} &= \frac{\partial b_{t+1}(i)}{\partial b_t(j)} \\
&= \frac{\partial}{\partial b_t(j)} \left[ \frac{\sum_{x \in S} b_t(x) T(i|x, a_t) O(z_t|a_t, i)}{\sum_{x \in S} \sum_{x' \in S} b_t(x) T(x'|x, a_t) O(z_t|a_t, x')} \right] \\
&= \frac{T(i|j, a_t) O(z_t|a_t, i)}{\sum_{x \in S} \sum_{x' \in S} b_t(x) T(x'|x, a_t) O(z_t|a_t, x')} \\
&\qquad\qquad - \frac{\sum_{x' \in S} T(x'|j, a_t) O(z_t|a_t, x')}{(\sum_{x \in S} \sum_{x' \in S} b_t(x) T(x'|x, a_t) O(z_t|a_t, x'))^2} \, .
\end{aligned}
$$

### A.2.1 PARTIAL DERIVATIVES

The derivative of $Q(\theta; \hat{\theta})$ with respect to $T(s'|s, a)$ is

$$
\begin{aligned}
\frac{\partial Q(\theta; \hat{\theta})}{\partial T(s'|s, a)} &= \frac{\partial}{\partial T(s'|s, a)} \sum_{i=1}^{n} \left[ \sum_{t=1}^{\tau} \mathbb{I}\{a_t = a\} \sum_{x \in S} \sum_{x' \in S} \xi_t(x, x') \log T(x'|x, a) \right. \\
&\qquad\qquad\qquad\qquad\qquad\qquad\qquad\qquad\qquad \left. + \sum_{t=2}^{\tau} \log \pi(a_t|b_t) \right] \\
&= \sum_{i=1}^{n} \left[ \sum_{t=1}^{\tau} \mathbb{I}\{a_t = a\} \frac{\xi_t(s, s')}{T(s'|s, a)} + \sum_{t=2}^{\tau} \frac{\partial \log \pi(a_t|b_t)}{\partial T(s'|s, a)} \right] \\
&= \sum_{i=1}^{n} \left[ \sum_{t=1}^{\tau} \mathbb{I}\{a_t = a\} \frac{\xi_t(s, s')}{T(s'|s, a)} \right. \\
&\qquad\qquad\qquad \left. + \sum_{t=2}^{\tau} \sum_{t'=1}^{t-1} \nabla_{b_t} \log \pi(a_t|b_t) \nabla_{b_{t'+1}} b_t \nabla_{T(s'|s, a)} b_{t'+1} \right] \, ,
\end{aligned}
$$

where

$$
\begin{aligned}
(\nabla_{b_t} \log \pi(a_t|b_t))_{1j} &= \frac{\partial \log(a_t|b_t)}{\partial b_t(j)} \\
&= \frac{\partial}{\partial b_t(j)} \left( -\eta \|b_t - \mu_{a_t}\|^2 - \log \sum_{a' \in A} e^{-\eta \|b_t - \mu_{a'}\|^2} \right) \\
&= -2\eta(b_t(j) - \mu_{a_t}(j)) + 2\eta \sum_{a \in A} \frac{e^{-\eta \|b_t - \mu_a\|^2}}{\sum_{a' \in A} e^{-\eta \|b_t - \mu_{a'}\|^2}} (b_t(j) - \mu_a(j)) \\
&= -2\eta(b_t(j) - \mu_{a_t}(j)) + 2\eta \sum_{a \in A} \pi(a|b_t)(b_t(j) - \mu_a(j))
\end{aligned}
$$

and

$$
\begin{aligned}
(\nabla_{T(s'|s, a)} b_{t'+1})_{i1} &= \frac{\partial b_{t'+1}(i)}{\partial T(s'|s, a)} \\
&= \frac{\partial}{\partial T(s'|s, a)} \left( \frac{\sum_{x \in S} b_{t'}(x) T(i|x, a_{t'}) O(z_{t'}|a_{t'}, i)}{\sum_{x \in S} \sum_{x' \in S} b_{t'}(x) T(x'|x, a_{t'}) O(z_{t'}|a_{t'}, x')} \right) \\
&= \mathbb{I}\{a_{t'} = a\} \left( \frac{\mathbb{I}\{i = s'\} b_{t'}(s) O(z_{t'}|a, s')}{\sum_{x \in S} \sum_{x' \in S} b_{t'}(x) T(x'|x, a) O(z_{t'}|a, x')} \right. \\
&\qquad\qquad\qquad \left. - \frac{b_{t'}(s) O(z_{t'}|a, s')}{(\sum_{x \in S} \sum_{x' \in S} b_{t'}(x) T(x'|x, a) O(z_{t'}|a, x'))^2} \right) \, .
\end{aligned}
$$

The derivative of $Q(\theta; \hat{\theta})$ with respect to $O(z|a, s')$ is

$$
\frac{\partial Q(\theta; \hat{\theta})}{\partial O(z|a, s')} = \frac{\partial}{\partial O(z|a, s')} \sum_{i=1}^{n} \left[ \sum_{t=1}^{\tau} \mathbb{I}\{a_t = a, z_t = z\} \sum_{x' \in S} \gamma_{t+1}(x') \log O(z|a, x') \right.
$$

$$+ \sum_{t=2}^{\tau} \log \pi(a_t | b_t) \Bigg]$$

$$= \sum_{i=1}^{n} \left[ \sum_{t=1}^{\tau} \mathbb{I}\{a_t = a, z_t = z\} \frac{\gamma_{t+1}(s')}{O(z|a, s')} + \sum_{t=2}^{\tau} \frac{\partial \log \pi(a_t | b_t)}{\partial O(z|a, s')} \right]$$

$$= \sum_{i=1}^{n} \left[ \sum_{t=1}^{\tau} \mathbb{I}\{a_t = a, z_t = z\} \frac{\gamma_{t+1}(s')}{O(z|a, s')} \right.$$

$$\left. + \sum_{t=2}^{\tau} \sum_{t'=1}^{t-1} \nabla_{b_t} \log \pi(a_t | b_t) \nabla_{b_{t'+1}} b_t \nabla_{O(z|a, s')} b_{t'+1} \right] ,$$

where

$$(\nabla_{O(z|a, s')} b_{t'+1})_{i1} = \frac{\partial b_{t'+1}(i)}{\partial O(z|a, s')}$$

$$= \frac{\partial}{\partial O(z|a, s')} \left( \frac{\sum_{x \in S} b_{t'}(x) T(i|x, a_{t'}) O(z_{t'}|a_{t'}, i)}{\sum_{x \in S} \sum_{x' \in S} b_{t'}(x) T(x'|x, a_{t'}) O(z_{t'}|a_{t'}, x')} \right)$$

$$= \mathbb{I}\{a_{t'} = a, z_{t'} = z\} \left( \frac{\mathbb{I}\{i = s'\} \sum_{x \in S} b_{t'}(x) T(s'|x, a)}{\sum_{x \in S} \sum_{x' \in S} b_{t'}(x) T(x'|x, a) O(z|a, x')} \right.$$

$$\left. - \frac{\sum_{x \in S} b_{t'}(x) T(s'|x, a)}{(\sum_{x \in S} \sum_{x' \in S} b_{t'}(x) T(x'|x, a) O(z|a, x'))^2} \right) .$$

The derivative of $Q(\theta; \hat{\theta})$ with respect to $b_1(s)$ is

$$\frac{\partial Q(\theta; \hat{\theta})}{\partial b_1(s)} = \frac{\partial}{\partial b_1(s)} \sum_{i=1}^{n} \left[ \sum_{x \in S} \gamma_1(x) \log b_1(x) + \sum_{t=1}^{\tau} \log \pi(a_t | b_t) \right]$$

$$= \sum_{i=1}^{n} \left[ \frac{\gamma_1(s)}{b_1(s)} + \sum_{t=1}^{\tau} \nabla_{b_t} \log \pi(a_t | b_t) \nabla_{b_1(s)} b_t \right] ,$$

where $(\nabla_{b_1(s)} b_t)_{i1} = (\nabla_{b_1} b_t)_{is}$.

The derivative of $Q(\theta; \hat{\theta})$ with respect to $\eta$ is

$$\frac{\partial Q(\theta; \hat{\theta})}{\partial \eta} = \frac{\partial}{\partial \eta} \sum_{i=1}^{n} \sum_{t=1}^{\tau} \log \pi(a_t | b_t)$$

$$= \sum_{i=1}^{n} \sum_{t=1}^{\tau} \frac{\partial}{\partial \eta} \left( -\eta \|b_t - \mu_{a_t}\|^2 - \log \sum_{a' \in A} e^{-\eta \|b_t - \mu_{a'}\|^2} \right)$$

$$= \sum_{i=1}^{n} \sum_{t=1}^{\tau} \left( -\|b_t - \mu_{a_t}\|^2 + \sum_{a \in A} \frac{e^{-\eta \|b_t - \mu_a\|^2}}{\sum_{a' \in A} e^{-\eta \|b_t - \mu_{a'}\|^2}} \|b_t - \mu_a\|^2 \right)$$

$$= \sum_{i=1}^{n} \sum_{t=1}^{\tau} \left( -\|b_t - \mu_{a_t}\|^2 + \sum_{a \in A} \pi(a|b_t) \|b_t - \mu_a\|^2 \right) .$$

Finally, the derivative of $Q(\theta; \hat{\theta})$ with respect to $\mu_a(s)$ is

$$\frac{\partial Q(\theta; \hat{\theta})}{\partial \mu_a(s)} = \frac{\partial}{\partial \mu_a(s)} \sum_{i=1}^{n} \sum_{t=1}^{\tau} \log \pi(a_t | b_t)$$

$$= \sum_{i=1}^{n} \sum_{t=1}^{\tau} \frac{\partial}{\partial \mu_a(s)} \left( -\eta \|b_t - \mu_{a_t}\|^2 - \log \sum_{a' \in A} e^{-\eta \|b_t - \mu_{a'}\|^2} \right)$$

$$= \sum_{i=1}^{n} \sum_{t=1}^{\tau} \left( 2\eta \mathbb{I}\{a_t = a\}(b_t(s) - \mu_a(s)) - 2\eta \frac{e^{-\eta \|b_t - \mu_a\|^2}}{\sum_{a' \in A} e^{-\eta \|b_t - \mu_{a'}\|^2}} (b_t(s) - \mu_a(s)) \right)$$

$$= \sum_{i=1}^{n} \sum_{t=1}^{\tau} (2\eta \mathbb{I}\{a_t = a\}(b_t(s) - \mu_a(s)) - 2\eta \pi(a|b_t)(b_t(s) - \mu_a(s))$$

$$= \sum_{i=1}^{n} \sum_{t=1}^{\tau} 2\eta (\mathbb{I}\{a_t = a\} - \pi(a|b_t))(b_t(s) - \mu_a(s)) .$$

# B  PROOFS OF PROPOSITIONS

## B.1  PROOF OF PROPOSITION 1

First, denote with $q_R^* \in \mathbb{R}^{\Delta(S) \times A}$ the optimal (belief-state) $q$-value function with respect to the underlying (state-space) reward function $R \in \mathbb{R}^{S \times A}$, and denote with $v^* \in \mathbb{R}^{\Delta(S)}$ the corresponding optimal value function $v_R^*(b) = \text{softmax}_{a' \in A} q_R^*(b, a')$. Now, fix some component $i$ of parameters $\theta$; we wish to compute the derivative of $\log \pi(a|b)$ with respect to $\theta_i$:

$$\frac{\partial}{\partial \theta_i} \log \pi(a|b) = \frac{\partial}{\partial \theta_i} \left( q_R^*(b, a) - v_R^*(b) \right)$$

$$= \frac{\partial}{\partial \theta_i} \left( q_R^*(b, a) - \log \sum_{a' \in A} e^{q_R^*(b, a')} \right)$$

$$= \frac{\partial}{\partial \theta_i} q_R^*(b, a) - \sum_{a' \in A} \left( \frac{e^{q_R^*(b, a')}}{\sum_{a'' \in A} e^{q_R^*(b, a'')}} \cdot \frac{\partial}{\partial \theta_i} q_R^*(b, a') \right)$$

$$= \frac{\partial}{\partial \theta_i} q_R^*(b, a) - \sum_{a' \in A} \pi(a'|b) \frac{\partial}{\partial \theta_i} q_R^*(b, a')$$

$$= \frac{\partial}{\partial \theta_i} q_R^*(b, a) - \mathbb{E}_{a' \sim \pi(\cdot|b)} \left[ \frac{\partial}{\partial \theta_i} q_R^*(b, a') \right]$$

where we make explicit here the dependence on $R$, but note that it is itself a parameter; that is, $R = \theta_j$ for some $j$. We see that this in turn requires computing the partial derivative $\partial q_R^*(b, a)/\partial \theta_i$. Let $\gamma$ be some appropriate discount rate, and denote with $\rho_R \in \mathbb{R}^{\Delta(S) \times A}$ the effective (belief-state) reward $\rho_R(b, a) \doteq \sum_{s \in S} b(s) R(s, a)$ corresponding to $R$. Further, let

$$\mathbb{P}(b'|b, a)$$

$$= \sum_{z \in Z} \mathbb{P}(z|b, a) \mathbb{P}(b'|b, a, z)$$

$$= \sum_{z \in Z} \left( \sum_{s \in S} \sum_{s' \in S} b(s) T(s'|s, a) O(z|a, s') \right) \delta \left( b' - \frac{\sum_{s \in S} b(s) T(\cdot|s, a) O(z|a, \cdot)}{\sum_{s \in S} \sum_{s' \in S} b(s) T(s'|s, a) O(z|a, s')} \right)$$

denote the (belief-state) transition probabilities induced by $T$ and $O$, where $\delta$ is the Dirac delta function such that $\delta(b')$ integrates to one if and only if $b' = \mathbf{0}$ is included in the integration region. Then the partial $\partial q_R^*(b, a)/\partial \theta_i$ is given as follows:

$$\frac{\partial}{\partial \theta_i} q_R^*(b, a) = \frac{\partial}{\partial \theta_i} \left( \rho_R(b, a) + \gamma \int_{b' \in \Delta(S)} \mathbb{P}(b'|b, a) v_R^*(b') db' \right)$$

$$= \underbrace{\frac{\partial}{\partial \theta_i} \rho_R(b, a) + \gamma \int_{b' \in \Delta(S)} v_R^*(b') \frac{\partial}{\partial \theta_i} \mathbb{P}(b'|b, a) db'}_{\rho_{R,i}(b, a)}$$

$$+ \gamma \int_{b' \in \Delta(S)} \mathbb{P}(b'|b, a) \mathbb{E}_{a' \sim \pi(\cdot|b')} \left[ \frac{\partial}{\partial \theta_i} q_R^*(b', a') \right] db'$$

from which we observe that $\partial q_R^*(b, a)/\partial \theta_i$ is a fixed point of a certain Bellman-like operator. Specifically, fix any function $f \in \mathbb{R}^{\Delta(S) \times A}$; then $\partial q_R^*(b, a)/\partial \theta_i$ is the fixed point of the operator

$\mathcal{T}_{R,i}^{\pi} : \mathbb{R}^{\Delta(S) \times A} \to \mathbb{R}^{\Delta(S) \times A}$ defined as follows:

$$(\mathcal{T}_{R,i}^{\pi} f)(b,a) = \rho_{R,i}(b,a) + \gamma \int_{b' \in \Delta(S)} \mathbb{P}(b'|b,a) \sum_{a'} \pi(a'|b') f(b',a') db'$$

which takes the form of a "generalized" Bellman operator on $q$-functions for POMDPs, where for brevity here we have written $\rho_{R,i}(b,a)$ to denote the expression $\frac{\partial}{\partial \theta_i} \rho_R(b,a) + \gamma \int_{b' \in \Delta(S)} v_R^*(b') \frac{\partial}{\partial \theta_i} \mathbb{P}(b'|b,a) db'$. Mathematically, this means that a recursive procedure can in theory be defined—cf. "$\nabla q$-iteration", analogous to $q$-iteration; see e.g. [52]—that may converge on the gradient under appropriate conditions. Computationally, however, this also means that taking a single gradient is at least as hard as solving POMDPs in general.

Further, note that while typical POMDP solvers operate by taking advantage of the convexity property of $\rho_R(b,a)$—see e.g. [53]—here there is no such property to make use of: In general, it is *not* the case that $\rho_{R,i}(b,a)$ is convex. To see this, consider the following counterexample: Let $S \doteq \{s_-, s_+\}$, $A \doteq \{a_=\}$, $Z \doteq \{z_-, z_+\}$, $T(s_-|s_-,a_=) = T(s_+|s_+,a_=) = p = 1$, $O(z_-|a_=,s_-) = O(z_+|a_=,s_+) = 1/4$, $b_1(s_+) = 1/2$, $R(s_-,a_=) = 0$, $R(s_+,a_=) = 1/2$, and $\gamma = 1/2$. For simplicity, we will simply write $b$ instead of $b(s_+)$. Note that:

$$q_R^*(b,a_=) = b \sum_{t=0} \gamma^t R(s_+,a_=) + (1-b) \sum_{t=0} \gamma^t R(s_-,a_=) = b$$

$$v_R^*(b) = \log \sum_{a \in \{a_=\}} e^{q_R^*(b,a)} = \log e^{q_R^*(b,a_=)} = q_R^*(b,a_=) = b$$

$$\mathbb{P}(z_+|b,a_=) = \frac{1}{4}(bp + (1-b)(1-p)) + \frac{3}{4}(b(1-p) + (1-b)p) = \frac{1}{4}b + \frac{3}{4}(1-b)$$

$$\mathbb{P}(z_-|b,a_=) = \frac{1}{4}(b(1-p) + (1-b)p) + \frac{3}{4}(bp + (1-b)(1-p)) = \frac{1}{4}(1-b) + \frac{3}{4}b$$

$$b'|b,a_=,z_+ = \frac{\mathbb{P}(s'=s_+, z_+|b,a_=)}{\mathbb{P}(z_+|b,a_=)} = \frac{\frac{1}{4}bp + \frac{3}{4}(1-b)(1-p)}{\mathbb{P}(z_+|b,a_=)} = \frac{\frac{1}{4}b}{\frac{1}{4}b + \frac{3}{4}(1-b)}$$

$$b'|b,a_=,z_- = \frac{\mathbb{P}(s'=s_+, z_-|b,a_=)}{\mathbb{P}(z_-|b,a_=)} = \frac{\frac{1}{4}(1-b)(1-p) + \frac{3}{4}bp}{\mathbb{P}(z_-|b,a_=)} = \frac{\frac{3}{4}b}{\frac{1}{4}(1-b) + \frac{3}{4}b}$$

$$\mathbb{P}(b'|b,a_=) = \begin{cases} \mathbb{P}(z_+|b,a_=) & \text{if } b' = b'|b,a_=,z_+ \\ \mathbb{P}(z_-|b,a_=) & \text{if } b' = b'|b,a_=,z_- \\ 0 & \text{otherwise} \end{cases}$$

Now, let the elements of $\theta$ be ordered such that $p$ is the $i$-th element, and consider $\rho_{R,i}(b,a)$—evaluated at $p = 1$:

$$\rho_{R,i}(b,a_=) \doteq \frac{\partial}{\partial p} \rho_R(b,a_=) + \gamma \int_{b' \in \Delta(S)} v_R^*(b') \frac{\partial}{\partial p} \mathbb{P}(b'|b,a_=) db'$$

$$= \frac{1}{2}\left( v_R^*(b'|b,a_=,z_+) \frac{\partial}{\partial p} \mathbb{P}(z_+|b,a_=) + v_R^*(b'|b,a_=,z_-) \frac{\partial}{\partial p} \mathbb{P}(z_-|b,a_=) \right)$$

$$= \frac{1}{2}\left( \frac{\frac{1}{4}b}{\frac{1}{4}b + \frac{3}{4}(1-b)} \left( \frac{1}{4}b - \frac{1}{4}(1-b) - \frac{3}{4}b + \frac{3}{4}(1-b) \right) \right.$$

$$\left. + \frac{\frac{3}{4}b}{\frac{1}{4}(1-b) + \frac{3}{4}b} \left( -\frac{1}{4}b + \frac{1}{4}(1-b) + \frac{3}{4}b - \frac{3}{4}(1-b) \right) \right)$$

Clearly $\rho_{R,i}(b,a_=)$ cannot be convex since $\rho_{R,i}(1/2, a_=) = 0$ and $\rho_{R,i}(1, a_=) = 0$ but $\rho_{R,i}(3/4, a_=) > 0$.

## B.2 Proof of Proposition 2

In contrast, unlike the indirect $q$-value parameterization above (which by itself requires approximate solutions to optimization problems), the mean-vector parameterization of INTERPOLE maps beliefs

directly to distributions over actions. Now, the derivatives of $\log \pi(a|b)$ are given as closed-form expressions in Appendices A.1 and A.2.

In particular, note that each $b_t$ is computed through a feed-forward structure, and therefore can easily be differentiated with respect to the unknown parameters $\theta$ through backpropagation through time: Each time step leading up to an action corresponds to a "hidden layer" in a neural network, and the initial belief corresponds to the "features" that are fed into the network; the transition and observation functions correspond to the weights between layers, the beliefs at each time step correspond to the activations between layers, the actions themselves correspond to class labels, and the action likelihood corresponds to the loss function (see Appendices A.1 and A.2).

Finally, note that computing all of the forward-backward messages $\alpha_t$ and $\beta_t$ in Appendix A.1 has complexity $O(n\tau S^2)$, computing all of the Jacobian matricies $\nabla_{b_t} b_{t'}$ in Appendix A.2 has complexity $O(n\tau^2 S^3)$, and computing all of the partial derivatives given in Appendix A.2 has complexity at most $O(n\tau^2 S^2 AZ)$. Hence, fully differentiating the expected log-likelihood $Q(\theta; \hat{\theta})$ with respect to the unknown parameters $\theta$ has an overall (polynomial) complexity $O(n\tau^2 S^2 \max\{S, AZ\})$.

## C EXPERIMENT PARTICULARS

### C.1 DETAILS OF DECISION ENVIRONMENTS

**ADNI** We have filtered out visits without a CDR-SB measurement, which is almost always taken, and visits that do not occur immediately after the six-month period following the previous visit but instead occur after 12 months or later. This filtering leaves 1,626 patients with typically three consecutive visits each. For MRI outcomes, average is considered to be within half a standard deviation of the population mean. Since there are only two actions in this scenario, we have set $\eta = 1$ and relied on the distance between the two means to adjust for the stochasticity of the estimated policy—closer means being somewhat equivalent to a smaller $\eta$.

**DIAG** We set $T^{\text{true}}(s_-|s_-, \cdot) = T^{\text{true}}(s_+|s_+, \cdot) = 1$, meaning patients do not heal or contract the diseases as the diagnosis progresses, $O^{\text{true}}(z_-|a_=, s_+) = O^{\text{true}}(z_+|a_=, s_-) = 0.4$, meaning measurements as a test have a false-negative and false-positive rates of $40\%$, and $b_1^{\text{true}}(s_+) = 0.5$. Moreover, the behavior policy is given by $T = T^{\text{true}}$, $O = O^{\text{true}}$, $b_1 = b_1^{\text{true}}$, $\eta = 10$, $\mu_{a_=}(s_+) = 0.5$, and $\mu_{a_-}(s_-) = \mu_{a_+}(s_+) = 1.3$. Intuitively, doctors continue monitoring the patient until they are $90\%$ confident in declaring a final diagnosis. In this scenario, $T$ and $\eta$ are assumed to be known. The behavior dataset is generated as 100 demonstration trajectories.

**BIAS** We set all parameters exactly the same way we did in DIAG with one important exception: now $O(s_-|a_=, z_+) = 0.2$ while it is still the case that $O^{\text{true}}(z_-|a_=, s_+) = 0.4$, meaning $O \neq O^{\text{true}}$ anymore. In this scenario, $b_1$ is also assumed to be known (in addition to $T$ and $\eta$) to avoid any invariances between $b_1$ and $O$ that we have encountered during training. The behavioral dataset is generated as 1000 demonstration trajectories.

### C.2 DETAILS OF BENCHMARK ALGORITHMS

**R-BC** We train an RNN whose inputs are the observed histories $h_t$ and whose outputs are the predicted probabilities $\hat{\pi}(a|h_t)$ of taking action $a$ given the observed history $h_t$. The network consists of an LSTM unit of size $64$ and a fully-connected hidden layer of size $64$. We minimize the cross-entropy loss $\mathcal{L} = -\sum_{i=1}^{n} \sum_{t=1}^{\tau} \sum_{a \in A} \mathbb{I}\{a_t = a\} \log \hat{\pi}(a|h_t)$ using Adam optimizer with learning rate 0.001 until convergence, that is when the cross-enropy loss does not improve for 100 consecutive iterations.

**PO-IRL** The IOHMM parameters $T$, $O$, and $b_1$ are initialized by sampling them uniformly at random. Then, they are estimated and fixed using conventional IOHMM methods. The reward parameter $R$ is initialized as $\hat{R}^0(s, a) = \varepsilon_{s,a}$ where $\varepsilon_{s,a} \sim \mathcal{N}(0, 0.001^2)$. Then, it is estimated via Markov chain Monte Carlo (MCMC) sampling, during which new candidate samples are generated by adding Gaussian noise with standard deviation 0.001 to the last sample. A final estimate is formed by averaging every tenth sample among the second set of 500 samples, ignoring the first 500 samples. In order to compute optimal q-values, we have used an off-the-shelf POMDP solver available at `https://www.pomdp.org/code/index.html`.

**Off. PO-IRL**   All parameters are initialized exactly the same way as in PO-IRL. Then, both the IOHMM parameters $T$, $O$, and $b_1$, and the reward parameter $R$ are estimated jointly via MCMC sampling. When generating new candidate samples, with equal probabilities, we have either sampled new $T$, $O$, and $b_1$ from IOHMM posterior (without changing $R$) or obtained a new $R$ the same way we did in PO-IRL (without changing $T$, $O$, and $b_1$). A final estimate is formed the same way as in PO-IRL.

**PO-MB-IL**   The IOHMM parameters $T$, $O$, and $b_1$ are initialized by sampling them uniformly at random. Then, they are estimated and fixed using conventional IOHMM methods. Given the IOHMM parameters, we parameterized policies the same way we did in INTERPOLE, that is as described in (2). The policy parameters $\{\mu_a\}_{a \in A}$ are initialized as $\hat{\mu}_a^0(s) = (1/|S| + \varepsilon_{a,s})/\sum_{s' \in S}(1/|S| + \varepsilon_{a,s'})$ where $\varepsilon_{a,s'} \sim \mathcal{N}(0, 0.001^2)$. Then, they are estimated according solely to the action likelihoods in (4) using the EM algorithm. The expected log-posterior is maximized using Adam optimizer with learning rate 0.001 until convergence, that is when the expected log-posterior does not improve for 100 consecutive iterations.

**INTERPOLE**   All parameters are initialized exactly the same way as in PO-MB-IL. Then, the IOHMM parameters $T$, $O$, and $b_1$, and the policy parameters $\{\mu_a\}_{a \in A}$ are estimated jointly according to both the action likelihoods and the observation likelihoods in (4). The expected log-posterior is again maximized using Adam optimizer with learning rate 0.001 until convergence.

### C.3   FURTHER EXAMPLE: POST-HOC ANALYSES

Policy representations learned by INTEPOLE provide users with means to derive concrete criteria that describe observed behavior in objective terms. These criteria, in turn, enable the quantitative analyses of the behavior using conventional statistical methods. For ADNI, we have considered two such criteria: *belatedness* of individual diagnoses and *informativeness* of individual tests. Both of these criteria are relevant to the discussion of early diagnosis, which is paramount for Alzheimer's disease [51] as we have already mentioned during the illustrative examples.

Formally, we consider the final diagnoses of a patient to be belated if (i) the patient was not ordered an MRI in one of their visits despite the fact that an MRI being ordered was the most likely outcome according to the policy estimated by INTERPOLE and (ii) the patient was ordered an MRI in a later visit that led to a near-certain diagnosis with at least $90\%$ confidence according to the underlying beliefs estimated by INTERPOLE. We consider An MRI to be uninformative if it neither (factually) caused nor could have (counterfactually) caused a significant change in the underlying belief-state of the patient, where an insignificant change is half a standard deviation less than the mean factual change in beliefs estimated by INTERPOLE.

Having defined belatedness and informativeness, one can investigate the frequency of belated diagnoses and uninformative MRIs in different cohorts of patients to see how practice varies between one cohort to another. In Table 5, we do so for six cohorts: all of the patients, patients who are over 75 years old, patients with apoE4 risk factor for dementia, patients with signs of MCI or dementia since their very first visit, female patients, and male patients. Note that increasing age, apoE4 allele, and female gender are known to be associated with increased risk of Alzheimer's disease [54–57]. For instance, we see that uninformative MRIs are much more prevalent among patients with signs of MCI or dementia since their first visit. This could potentially be because these patients are monitored much more closely than usual given their condition.

Table 5: Frequency of belated diagnoses and uninformative MRIs in various patient cohorts.

| Cohort | Frequency of belated diagnoses | Frequency. of uninformative MRIs |
|---|---|---|
| All patients | 6.52% | 18.8% |
| Patients over 75 years old | 9.29% | 18.1% |
| Patients with apoE4 risk factor | 8.75% | 19.3% |
| Patients with signs of MCI/dementia | 9.26% | 26.1% |
| Female patients | 7.19% | 17.6% |
| Male patients | 5.97% | 19.8% |

Alternatively, one can divide patients into cohorts based on whether they have a belated diagnoses or an uninformative MRI to see which features these criteria correlate with more. We do so in Table 6. For instance, we see that a considerable percentage of belated diagnoses are seen among male patients.

Table 6: Features of patients with belated diagnoses and uninformative MRIs.

| Feature | All patients | Patients with belated diagnoses | Patients with uninformative MRIs |
|---|---|---|---|
| Mean age | $73.9 \pm 7.1$ | $75.8 \pm 7.5$ | $73.0 \pm 7.3$ |
| Freq. of apoE4 | 45.7% | 54.2% | 49.0% |
| Freq. of MCI/dementia signs | 68.4% | 95.8% | 98.1% |
| Perc. of female patients | 45.4% | 39.0% | 43.5% |
| Perc. of male patients | 54.6% | 61.0% | 56.5% |

## C.4 FURTHER EXAMPLE: DECISION TREES

Clinical practice guidelines are often given in the form of decision trees, which usually have vague elements that require the judgement of the practitioner [58, 59]. For example, the guideline could ask the practitioner to quantify risks, side effects, or improvements in subjective terms such as being significant, serious, or potential. Using direct policy learning, how vague elements like these are commonly resolved in practice can be learned in objective terms.

Formulating policies in terms of IOHMMs and decision boundaries is expressive enough to model decision trees. An IOHMM with deterministic observations, that is $O(z|a, s') = 1$ for some $z \in Z$ and for all $a \in A, s \in S$, essentially describes a finite-state machine, inputs of which are equivalent to the observations. Similarly, a deterministic decision tree can be defined as a finite-state machine with no looping sequence of transitions. The case where the observations are probabilistic rather than deterministic correspond to the case where the decision tree is traversed in a probabilistic way so that each path down the tree has a probability associated with it at each step of the traversal.

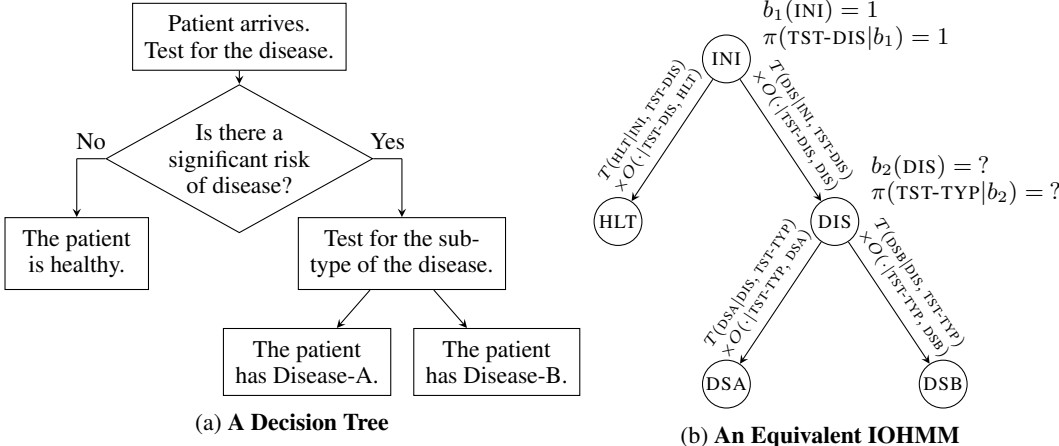

(a) **A Decision Tree**

(b) **An Equivalent IOHMM**

Figure 5: *Two Different Descriptions of the Same Policy:* (a) in the form of a decision tree and (b) in terms of an equivalent IOHMM. For the IOHMM in (b), arrows denote possible transitions, where the probability of a transition is proportional to the quantity written above the corresponding arrow. Using direct policy learning, we can infer the risk of disease, $b_2(\text{DIS})$, and the probability of testing for the sub-type based on the risk, $\pi(\text{TST-TYP}|b_2)$, which are left vague in (a).

As a concrete example of modeling decision trees in terms of IOHMMs, consider the scenario of diagnosing a disease with two sub-types: Disease-A and Disease-B. Figure 5a depicts the policy of the doctors in the form of a decision tree. Each newly-arriving patient is first tested for the disease in a general sense without any distinction between the two sub-types it has. The patient is then tested

for a specific sub-type of the disease only if the doctors deem there is a significant risk that the patient is diseased. Note that which exact level of confidence constitutes as a significant risk is left vague in the decision tree. By modeling this scenario using our framework, we can learn: (i) how the risk is determined based on initial test results and (ii) what amount of risk is considered significant enough to require a subsequent test for the sub-type.

Let $S = \{\text{INI}, \text{HLT}, \text{DIS}, \text{DSA}, \text{DSB}\}$, where INI denotes that the patient has newly arrived, HLT denotes that the patient is healthy, DIS denotes that the patient is diseased, DSA denotes that the patient has Disease-A, and DSB denotes that the patient has Disease-B. Figure 5b depicts the state space $S$ with all possible transitions. Note that the initial belief $b_1$ is such that $b_1(\text{INI}) = 1$. Let $A = \{\text{TST-DIS}, \text{TST-TYP}, \text{STP-HLT}, \text{STP-DSA}, \text{STP-DSB}\}$, where TST-DIS denotes testing for the disease, TST-TYP denotes testing for the sub-type of the disease, and the remaining actions denote stopping and diagnosing the patient with one of the terminal states, namely states HLT, DSA, and DSB.

After taking action $a_1 = \text{TST-DIS}$ and observing some initial test result $z_1 \in Z$, the risk of disease, which is the probability that the patient is diseased, can be calculated with a simple belief update:

$$b_2(\text{DIS}) \propto \sum_{s \in S} b_1(s) T(\text{DIS}|s, \text{TST-DIS}) O(z_1|\text{TST-DIS}, \text{DIS})$$
$$= T(\text{DIS}|\text{INI}, \text{TST-DIS}) O(z_1|\text{TST-DIS}, \text{DIS}) \ .$$

Moreover, we can say that the doctors are more likely to test for the sub-type of the disease as opposed to stopping and diagnosing the patient as healthy, that is $\pi_b(\text{TST-TYP}|b_2) > \pi_b(\text{STP-HLT}|b_2)$, when

$$b_2(\text{DIS}) > \frac{\mu_{\text{TST-TYP}}(\text{DIS}) + \mu_{\text{STP-HLT}}(\text{DIS})}{2}$$

assuming $\mu_{\text{TST-TYP}}(\text{DIS}) > \mu_{\text{STP-HLT}}(\text{DIS})$. Note that there are only two possible actions at the second time step: actions TST-TYP and STP-HLT.

## D  DETAILS OF THE CLINICIAN SURVEYS

Each participant was provided a short presentation explaining (1) the ADNI dataset and the decision-making problem we consider, (2) what rewards and reward functions are, (3) what beliefs and belief simplices are, and (4) how policies can be represented in terms of reward functions as well as decision boundaries. Then, they were asked two multiple-choice questions, one that is strictly about representing histories, and one that is strictly about representing policies. Importantly, the survey was conducted *blindly*—i.e. they were given no context whatsoever as pertains this paper and our proposed method. The question slides can be found in Figures 6 and 7. Essentially, each question first states a hypothesis and shows two/three representations relevant to the hypothesis stated. Then, the participant is asked which of the representations shown most readily expresses the hypothesis. Here are the full responses that we have received, which includes some additional feedback:

- **Clinician 1**
  *Question 1*: C > B > A
  *Question 2*: B
  *Additional Feedback*: The triangle was initially more confusing than not, but the first example (100% uncertainty) was helpful. It isn't clear how the dots in the triangle are computed. Are these probabilities based on statistics? Diagram is always better than no diagram.

- **Clinician 2**
  *Question 1*: C > B > A
  *Question 2*: B
  *Additional Feedback*: I always prefer pictures to tables, they are much easier to understand.

- **Clinician 3**
  *Question 1*: C > B > A
  *Question 2*: B
  *Additional Feedback*: Of course the triangle is more concise and easier to look at. But how is the decision boundary obtained? Does the decision boundary always have to be parallel to one of the sides of the triangle?

- **Clinician 4**
  *Question 1*: C > B > A
  *Question 2*: B
  *Additional Feedback*: [Regarding Question 1,] representation A and B do not show any interpretation of the diagnostic test results, whereas representation C does. I think doctors are most familiar with representation B, as it more closely resembles the EHR. Although representation C is visually pleasing, I'm not sure how the scale of the sides of the triangle should be interpreted. [Regarding Question 2,] again I like the triangle, but it's hard to interpret what the scale of the sides of the triangle mean. I think option A is again what doctors are more familiar with.

- **Clinician 5**
  *Question 1*: C > B > A
  *Question 2*: A

- **Clinician 6**
  *Question 1*: C > B > A
  *Question 2*: A

- **Clinician 7**
  *Question 1*: C
  *Question 2*: B
  *Additional Feedback*: I thought I'd share with you my thoughts on the medical aspects in your scenario first (although I realise you didn't ask me for them). [...] The Cochrane review concludes that MRI provides low sensitivity and specificity and does not qualify it as an add on test for the early diagnosis due to dementia (Lombardi G et. al. Cochrane database 2020). The reason for MRI imaging is (according to the international guidelines) to exclude non-degenerative or surgical causes of cognitive impairment. [...] In your example the condition became apparent when the CDR-SB score at Month 24 hit 3.0 (supported by the sequence of measurements over time showing worsening CDR-SB score). I imagine the MRI was triggered by slight worsening in the CDR-SB score (to exclude an alternative diagnosis). To answer your specific questions: Q1. The representation C describes your (false) hypothesis that it was the MRI that made the diagnosis of MCI more likely/apparent the best—I really like the triangles. Q2. I really like the decision boundary.

- **Clinician 8**
  *Question 1*: C > B > A
  *Question 2*: B

- **Clinician 9**
  *Question 1*: C
  *Question 2*: B
  *Additional Feedback*: Q1. Representation C gives the clearest illustration of the diagnostic change following MRI. However, the representation of beliefs on a continuous spectrum around discrete cognitive states could be potentially confusing given that cognitive function is itself a continuum (and 'MCI', 'Dementia' and 'NL' are stations on a spectrum rather than discrete states). Also, while representation C is the clearest illustration, it is the representation that conveys the least actual data and it isn't clear from the visualisation exactly what each shift in 2D space represents. Also, the triangulation in 'C' draws a direct connection between NL and Dementia, implying that this is a potential alternative route for disease progression, although this is more intuitively considered as a linear progression from NL to MCI to Dementia. Q2. For me, the decision boundary representation best expresses the concept of the likelihood of ordering and MRI with the same caveats described above. Option B does best convey the likelihood of ordering an MRI, but doesn't convey the information value provided by that investigation. However, my understanding is that this is not what you are aiming to convey here.

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
