# OpenReview forum: "Explaining by Imitating: Understanding Decisions by Interpretable Policy Learning"
_ICLR.cc/2021/Conference — ICLR 2021 Poster_

### Official Review · AnonReviewer1 · 2020-10-28
**Simple and effective idea but missing a key related work**

**Rating:** 6
**Confidence:** 3

**Review:**

The paper proposes a method for obtaining an interpretable representation of some behavioral policy based on partial observations and in an offline learning setting. The paper is well written and the approach is clearly explained. I just have few comments/questions:

In terms of alternative approaches to modeling agent/decision maker behavior, I can think of at least one other alternative and I wonder how your method can be compared to it: ‘Agent Markov Model’ introduced by Unhelkar and Shah (Learning Models of Sequential Decision-Making with Partial Specification of Agent Behavior AAAI 2019) is also modeling agent behavior using a state space model with partial observations. Similar to your approach, they are also interested in modeling the agent and not the actual mechanics of the world. As far as I understand AMM also satisfies three key criteria presented in the paper. AMM also uses a simpler model for the policy; they assume agent follows a stationary Markov policy: \pi(a|s, z). Can you discuss advantages of your model over theirs?

Another advantage of AMM over INTERPOLE is that it can infer the number of latent states via a nonparametric Bayesian approach. In INTERPOLE, do you have any recommendation on how S can be chosen in a more general setting? How sensitive are the results w.r.t model misspecification (specifically w.r.t. the number of latent state)?

---

> ### Author Response · Authors · 2020-11-19
> **Response to Reviewer #1 [Part 1/1]**
>
> Thank you for your thoughtful comments and suggestions. We give answers to each in turn, as well as pointing out corresponding updates to the revised manuscript (additions are indicated in *blue*).
>
> ---
>
> **(1) Comparison with AMMs**
>
> We thank the reviewer for pointing out AMMs. Indeed, their methodology bears some resemblance to ours and, in the revised version of the paper, we have now added a discussion of them to the related work.
>
> However, there are some key differences between AMMs and our model, which means AMMs do not fulfill some of the three key criteria that motivated the design of our approach.
>
> First to establish notation: AMMs model transitions from state $f=[x,s]$ to a new state $f’=[x’,s’]$ via the expression $T_s(s’|s,a)T_x(x’|x,s,a)$. Here, $s$ denotes the (objective) state of the environment and $T_s$ denotes the true environment dynamics whereas $x$ denotes the (subjective) internal state of the decision-maker, which would correspond to beliefs in our work, and $T_x$ denotes the internal update procedure of the decision-maker, which is equivalent to decision dynamics.
>
> (a) No Partial Observability: Now, note that AMMs assume that the environment is *fully observable*, and that the decision-maker updates their internal state $x$ on the basis of fully-observed states $s$. In our setting, the environment is partially observable and the beliefs are updated on the basis of partial observations. AMMs do not fulfill the partial-observability criterion.
>
> (b) Not Operable Offline: Moreover, the methodology used in Unhelkar and Shah (2019) assumes $T_s$ (i.e. the true environment dynamics) to be *known* when inferring $T_x$. Therefore this does not fulfill our key criterion that the method be operable offline (equivalently, operable in environments where the “true” dynamics are unknown).
>
> ---
>
> **(2) Number of Latent States**
>
> Allow us to reiterate a subtle distinction regarding state spaces. First, there may be some “ground-truth” external state space: This may indeed be arbitrarily complex or even impossible to discover, but---as explained throughout---we are actually not interested in modeling this. Second, there is the *internal* state space that an agent uses to reason about decisions: This is what we are interested in. In fact, our focus on *subjective* decision dynamics (instead of “true” environment dynamics) is what sets us apart from comparable literature.
>
> So the relevant question here is: Do we know the dimensionality of the (subjective) state space of a human agent? The answer is: very often we do---and the medical literature provides clear and ample evidence: The progression of many diseases, including dementia [42, 43], lung and breast cancer [44, 45], cystic fibrosis [46], and pulmonary disease [47] are often modeled in terms of three or four well-defined states. Since our goal is to obtain interpretable representations of decisions, it is therefore reasonable to cater precisely to these accepted state spaces that doctors can most readily reason with. We believe that describing behavior in terms of beliefs over these (already well-understood) states is one of the main contributors to the interpretability of our method.
>
> (Of course, there is also the---more conventional---question of what the “true” state space looks like, and whether we can model that as a scientific question. However, there are many existing methods to do that, and is outside of the scope of our focus.)
>
> Finally, it is true that the state spaces as represented in medical knowledge are often relatively simple, and likely do not correspond 100% to “true” environment dynamics. However, we reiterate that this discrepancy is *precisely* why we hope to model the subjective dynamics directly, instead of the environment dynamics (which are likely more complex, but not interpretable as they do not correspond to how doctors reason, for instance).
>
> We agree that the above distinction would benefit from further clarification. In the revised manuscript, we now include a version of the above explanation in the discussion section of the paper.

---

> ### Author Response · Authors · 2020-11-23
> **Dear Reviewer #1**
>
> Once again, thank you for your invaluable feedback. We were wondering whether our response and the revised manuscript addressed your concerns. If you have any additional comments, please let us know, we would be happy to address them.

---

### Official Review · AnonReviewer2 · 2020-10-28

**Rating:** 7
**Confidence:** 3

**Review:**

The paper proposes an algorithm for learning policies and internal models ("decision dynamics") from demonstrations. The key idea is to fit a distribution over policies, observation models, and transition models using an EM-like method. Offline experiments on a healthcare dataset show that the method learns interpretable decision dynamics, recovers biased internal models, and accurately predicts actions relative to prior methods.

Overall, the paper is well-written, the experiments are convincing, and the proposed method, Interpole, makes a meaningful contribution to the literature on modeling decision-making.

It would be nice to evaluate Interpole on tasks in which the demonstrator's decision dynamics operate on high-dimensional latent belief states, to illustrate how Interpole scales with the complexity of the demonstrator's internal models. In such settings, would the mean-vector representation in Equation 2 become problematic due to the curse of dimensionality?

There is some missing related work on learning from demonstrations that also satisfies the three criteria of transparency by design, partial observability, and offline learning: assistive state estimation [1], learning sensor models from demonstrations in the LQG setting [2], learning from a demonstrator with false beliefs [3], and inverse rational control [4].

1. https://arxiv.org/pdf/2008.02840.pdf
2. https://fias.uni-frankfurt.de/~rothkopf/docs/Schmitt_et_al_2017.pdf
3. https://arxiv.org/pdf/1512.05832.pdf
4. https://arxiv.org/pdf/2009.12576.pdf

Typos:
 - "log-like-lihood" -> log-likelihood (page 5)

---

> ### Author Response · Authors · 2020-11-19
> **Response to Reviewer #2 [Part 3/3]**
>
> **(3) Importance of Transparency**
>
> The primary difference between InterPole and [3] is that they recover *reward functions* (akin to the IRL approach, and in this sense similar to [2] and [4]), whereas our central thesis is that *decision dynamics/boundaries* are more interpretable.
>
> Given this, we posit that the most direct way to evaluate the “interpretability” of our method is simply to consult domain experts (i.e. clinicians in our case) for feedback. We therefore took the time to conduct such a study for the ADNI setting, and have now included our initial findings in the revised manuscript of the paper. The exact details of the new results can be found in the revised paper, but briefly, we have evaluated two aspects of interpretability regarding our method:
>
> (a) Decision Dynamics: Whether the proposed representation of (possibly subjective) belief trajectories are preferable to raw action-observation trajectories in terms of transparency---that is, whether decision dynamics are an interpretable way of modeling how information is aggregated by the decision-maker---and
>
> (b) Decision Boundaries: Whether the proposed representation of (possibly suboptimal) decision boundaries are a more transparent way of describing policies compared with the notion of reward functions, which is the conventional approach in the policy learning literature.
>
> We have reached out to nine clinicians from four different countries (United States, United Kingdom, the Netherlands, and China), asking them what is most preferable in terms of understandability, using the ADNI environment. Importantly, the survey was conducted blindly---i.e. they were given no context whatsoever as pertains this paper and our proposed method.
>
> - For the first aspect, we presented to them the medical history of an example patient represented in three ways: Using: (a) only the most recent action-observation, (b) the entire trajectory of actions and observations, and (c) belief trajectories as recovered by our method. Result: All nine clinicians preferred belief trajectories over action-observation trajectories.
>
> - For the second aspect, we showed them the policies learned by both Off. PO-IRL and Interpole, which parameterizes policies in terms of (a) reward functions and (b) decision boundaries respectively. Result: Seven out of nine clinicians preferred the representation in terms of decision boundaries over that of reward functions.
>
> We believe that these results provide strong evidence of the relative interpretability of our proposed approach---especially compared with “reward” functions.

---

> ### Author Response · Authors · 2020-11-19
> **Response to Reviewer #2 [Part 2/3]**
>
> **(2) Additional Related Work**
>
> We thank the reviewer for bringing [1,2,3,4] to our attention. We agree that they are related to our own work as all of them consider the subproblem of modeling a decision-maker’s internal recognition model (i.e. the model with which they reason about the observations they make). In the revised version, we have now added a discussion of [1,2,3,4] to the related work.
>
> However, we would like to point out that [1], [2], and [4] study completely different (main) problems than ours (without always adhering to the three key criteria that we have established) and [3], while tackling the same problem as we do, does not satisfy the transparency criterion.
>
> - [1] aims to generate “calibrated” observations for the decision-maker so that they behave optimally when they act on the basis of these “calibrated” observations. Doing so requires estimating the belief update process of the decision-maker (cf. decision dynamics). However, [1] assumes access to the true belief update procedure (cf. true dynamics) and the reward function of the decision-maker (cf. decision boundaries). Note that this is a completely different problem instance than ours; assuming the reward function of the decision-maker to be known is completely at odds with our goal of learning complete policies from scratch. Moreover, describing policies via reward functions violates the transparency criterion while having access to the true belief update procedure (i.e. knowing the true dynamics) violates the offline-operation criterion.
>
> - [2] and [4] study the problem of estimating the observations of an agent (as well as how the agent uses those observations to update their beliefs) given the fully-observed states of the environment and the actions of the agent. In this framework, what the learner sees and what the agent sees are not aligned. While the environment is fully-observable from the perspective of the agent, it is perceived as a partially-observable one by the decision-maker. However, this is not the case in medical settings, where neither our method nor the medical practitioner have access to the underlying (unobserved) state of the environment. We seek to infer an agent’s belief about never-observed states based on their (known) observations while [2] and [4] seek to infer an agent’s (unknown) observations (and the associated beliefs about those observations) based on fully-observed states. This alternative framework not only studies a completely different problem than ours, it also clearly violates the partial-observability criterion as we have defined it.
>
> - [3] directly tackles the same problem as we do. Moreover, unlike many existing work, it does not assume beliefs to be unbiased or policies to be optimal (in particular, non-optimal policies are modeled via time-inconsistent agents), which makes it significant from the perspective of our methodology. However, it still relies on rewards to model policies, which comes with all the drawbacks of reward-based models discussed in the apprenticeship learning paragraph of the related work section. Due to this, [3] violates the transparency criteria.

---

> ### Author Response · Authors · 2020-11-19
> **Response to Reviewer #2 [Part 1/3]**
>
> Thank you for your thoughtful comments and suggestions. We give answers to each in turn, as well as pointing out corresponding updates to the revised manuscript (additions are indicated in *blue*).
>
> ---
>
> **(1) Complexity of State Space**
>
> Allow us to reiterate a subtle distinction regarding state spaces. First, there may be some “ground-truth” external state space: This may indeed be arbitrarily complex or even impossible to discover, but---as explained throughout---we are actually not interested in modeling this. Second, there is the *internal* state space that an agent uses to reason about decisions: This is what we are interested in. In fact, our focus on *subjective* decision dynamics (instead of “true” environment dynamics) is what sets us apart from comparable literature.
>
> So the relevant question here is: How high-dimensional is the (subjective) state space of a human agent? The answer is: not high at all, and the medical literature provides clear and ample evidence: The progression of many diseases, including dementia [42, 43], lung and breast cancer [44, 45], cystic fibrosis [46], and pulmonary disease [47] are often modeled in terms of three or four well-defined states. Since our goal is to obtain interpretable representations of decisions, it is therefore reasonable to cater precisely to these accepted state spaces that doctors can most readily reason with. We believe that describing behavior in terms of beliefs over these (already well-understood) states is one of the main contributors to the interpretability of our method.
>
> (Of course, as a purely mathematical question, whether Algorithm 1 scales to high-dimensional state spaces is certainly a valid question. However, this is not what it was designed for, and is beyond the scope of our contribution).
>
> We agree that the above distinction would benefit from further clarification. In the revised manuscript, we now include a version of the above explanation in the discussion section of the paper.

---

> ### Author Response · Authors · 2020-11-23
> **Dear Reviewer #2**
>
> Once again, thank you for your invaluable feedback. We were wondering whether our response and the revised manuscript addressed your concerns. If you have any additional comments, please let us know, we would be happy to address them.

---

### Official Review · AnonReviewer3 · 2020-10-28

**Rating:** 7
**Confidence:** 3

**Review:**

Summary:
This work proposes an approach for understanding and explaining decision-making behavior. The authors aim to make the method 1) transparent, 2) able to handle partial observability, and 3) work with offline data. To do this, they develop INTERPOLE, which uses Bayesian techniques to estimate decision dynamics as well as decision boundaries. Results on simulated and real-world domains show that their method explains the decisions in behavior data while still maintaining accuracy and focuses on explaining decision dynamics rather than the “true” dynamics of the world.

Strengths:
- This work tackles an interesting problem. The proposed setting that considers interpretability + partial observability + offline data is important and reflective of many real-world problems so an approach that works in this setting is very meaningful.
- The paper is well-written and clear. The authors do a good job clearly stating the motivation for the work and differences from prior work.
- The authors consider both simulated and real-world data. The application to healthcare is useful and interesting. Overall, the evaluation helps support the claims, except for the interpretability component described below.

Weaknesses:
- My biggest concern with the work is that it argues for interpretability but the best way to evaluate interpretability is through human evaluation. I appreciate that the authors include the visualizations and talk through what each point means, but it wasn’t 100% clear whether it was truly interpretable. The best way to judge this would be to compare non-interpretable and interpretable techniques and have humans blindly rate which made more sense.

Other comments:
- In Section 3 under Learning Objective, is it reasonable to assume access to the state space?
- In Section 5, you talk about counterfactual updates. Can you describe this in more detail? This wasn’t super clear to me.
- Adding to the weakness point above, the visualization in Figure 3 does not fully make sense. I understand the points and the arrows, but it’s not very clear whether the point’s exact location in the belief simplex is meaningful (e.g., in the middle vs on the line between MCI and dementia).
- It looks like the action matching metric is used to evaluate the ability to imitate and the belief/policy mismatch is used for evaluating explainability. But belief/policy mismatch cannot be used for ADNI, so for this domain, there’s no proper evaluation of explainability other than the visualizations. I think this is a key piece that needs to be evaluated and shown.

Recommendation:
Overall, the work poses an interesting problem and solution, but the key motivation is to develop an interpretable approach, which I don’t think is sufficiently and correctly evaluated. Thus, I’m on the fence and would like to hear from the authors about this point.

=========================

Response after author rebuttal:
The authors answered my concern about evaluating the interpretability of the approach. They evaluated the method with a few clinicians, and I'm glad to see that they preferred the authors' method.

Adding these results + clarifying the points I included will definitely make the paper stronger. I increase my score as a result and recommend acceptance.

---

> ### Author Response · Authors · 2020-11-19
> **Response to Reviewer #3 [Part 2/2]**
>
> **(3) Counterfactual Updates**
>
> Thank you for pointing this out; we agree that this terminology would benefit from clarification.
>
> - “Factual update”: In Figure 3d, when the third belief is updated to obtain the fourth and the final belief, this is done according to Equation 1 given the third action $a_3$ taken and the third observation $z_3$ made. We call this a “factual” belief update in the sense that action $a_3$ and observation $z_3$ are the actual action and observation recorded in the dataset.
>
> - “Counterfactual update”: However, having access to an estimated model of the decision dynamics, we can also compute (using the same equation) alternative beliefs for alternative observations that were not actually made, but could have been made (i.e. for $z’_3\neq z_3$). This is what we have done when computing the “counterfactual” update in Figure 3d.
>
> Note that, when deciding on whether to take action $a_3$ or not, the decision-maker does not know what will be the resulting observation $z_3$. Hence, both the factual and counterfactual updates are possible outcomes before action $a_3$ is taken. Now, our key point for Figure 3d is simply as follows: Although the belief update that was realized (the factual one) seems uninformative, note that the ordered MRI *could have* resulted in a significant update (the counterfactual one) if observation $z_3$ were to be different.
>
> We have now added a clear description of what a counterfactual update is to the paper.
>
> ---
>
> **(4) Interpretation of Belief Simplex**
>
> We agree that Figure 3 would benefit from a more detailed explanation. In fact, the *belief simplex* is a standard and commonly-used representation of probability distributions, with a very natural interpretation.
>
> Each point in the simplex corresponds to a unique belief (i.e. probability distribution over the state space---which in this case is NL, MCI, and Dementia). The closer the point is to a vertex (i.e. state), the higher the probability assigned to that state. For instance:
>
> - If the belief is exactly in the middle of the simplex (i.e. equidistant from all vertices), then all states are believed to be equally likely (i.e. with probabilities 1/3, 1/3, 1/3).
>
> - If the belief is exactly on the line midway between MCI and Dementia, then this is very different from before: The probability assigned to NL is now virtually zero (i.e. the decision-maker has “ruled out” NL as the underlying state), and the probability assigned to MCI and Dementia is now 1/2 and 1/2.
>
> - Finally, if the belief is exactly on a vertex (e.g. directly on top of MCI), then this corresponds to an absolutely certain belief that the underlying state is MCI, with zero probability assigned to NL or Dementia.
>
> In the revised manuscript, we have now included this explanation to the interpretability subsection in illustrative examples.
>
> ---
>
> **(5) Evaluation of Explainability for ADNI**
>
> We agree that for ADNI further evaluation of interpretability is beneficial. Please kindly refer to our response (1) above, in which this point is addressed in detail.

---

> > ### Comment · AnonReviewer3 · 2020-11-25
> > **Response to authors**
> >
> > Thank you very much for your response. My biggest concern was on the interpretability evaluation. Thank you for evaluating this with a few clinicians, and I'm glad to see that they preferred your method.
> >
> > Adding these results + clarifying the points above will definitely make the paper stronger. I increase my score as a result.

---

> ### Author Response · Authors · 2020-11-19
> **Response to Reviewer #3 [Part 1/2]**
>
> Thank you for your thoughtful comments and suggestions. We give answers to each in turn, as well as pointing out corresponding updates to the revised manuscript (additions are indicated in *blue*).
>
> ---
>
> **(1) Human Evaluation**
>
> We agree that the most direct way to evaluate the “interpretability” of our method is simply to consult domain experts (i.e. clinicians in our case) for feedback. We therefore took the time to conduct such a study for the ADNI setting, and have now included our initial findings in the revised manuscript of the paper. The exact details of the new results can be found in the revised paper, but briefly, we have evaluated two aspects of interpretability regarding our method:
>
> (a) Decision Dynamics: Whether the proposed representation of (possibly subjective) belief trajectories are preferable to raw action-observation trajectories in terms of transparency---that is, whether decision dynamics are an interpretable way of modeling how information is aggregated by the decision-maker---and
>
> (b) Decision Boundaries: Whether the proposed representation of (possibly suboptimal) decision boundaries are a more transparent way of describing policies compared with the notion of reward functions, which is the conventional approach in the policy learning literature.
>
> We have reached out to nine clinicians from four different countries (United States, United Kingdom, the Netherlands, and China), asking them what is most preferable in terms of understandability, using the ADNI environment. Importantly, the survey was conducted blindly---i.e. they were given no context whatsoever as pertains this paper and our proposed method.
>
> - For the first aspect, we presented to them the medical history of an example patient represented in three ways: Using: (a) only the most recent action-observation, (b) the entire trajectory of actions and observations, and (c) belief trajectories as recovered by our method. Result: All nine clinicians preferred belief trajectories over action-observation trajectories.
>
> - For the second aspect, we showed them the policies learned by both Off. PO-IRL and Interpole, which parameterizes policies in terms of (a) reward functions and (b) decision boundaries respectively. Result: Seven out of nine clinicians preferred the representation in terms of decision boundaries over that of reward functions.
>
> We believe that these results provide strong evidence of the relative interpretability of our proposed approach.
>
> ---
>
> **(2) Access to State Space**
>
> Allow us to reiterate a subtle distinction. First, there may be some “ground-truth” external state space: This may be arbitrarily complex or even impossible to discover, but---as explained throughout---we are not interested in modeling this. Second, there is the *internal* state space that an agent uses to reason about decisions: This is what we are interested in.
>
> In this sense, it is certainly reasonable to assume access to the state space, which is often very clear from medical literature. The progression of many diseases, including dementia [42, 43], lung and breast cancer [44, 45], cystic fibrosis [46], and pulmonary disease [47] are often modeled in terms of three or four well-defined states. Since our goal is to obtain interpretable representations of decisions, it is therefore reasonable to cater precisely to these accepted state spaces that doctors can most readily reason with. We believe that describing behavior in terms of beliefs over these (already well-understood) states is one of the main contributors to the interpretability of our method.
>
> We agree that this distinction would benefit from further clarification. In the revised manuscript, we now include a version of the above explanation in the discussion section of the paper.

---

> ### Author Response · Authors · 2020-11-23
> **Dear Reviewer #3**
>
> Once again, thank you for your invaluable feedback. We were wondering whether our response and the revised manuscript addressed your concerns. If you have any additional comments, please let us know, we would be happy to address them.

---

### Decision · Program_Chairs · 2021-01-07
**Final Decision**

**Decision:**

Accept (Poster)

**Comment:**

Explaining by Imitating: Understanding Decisions by Interpretable
Policy Learning

The topic is maximally timely and important: Understanding human
decision-making behaviour based on observational data. Any tangible
steps towards this challenging goal are bound to be significant, and
those this paper makes.

A Bayesian policy-learning method is introduced for this task, and
validated on both simulated data and user exeperiments in a real
decision-making task. The novel contribution is on learning
interpretable decision dynamics

The paper is written clearly enough..

The updated paper clarified most major concerns the reviewers had. In
particular, they added a user study.

The biggest remaining weaknesses are that

- relationship to the AMM model did not become completely clear yet

- the real user study has been carried out with only a small set of
users. But a large-cohort study would be too much work to ask for a
paper which has also a strong methodological contribution.